# Uniform convergence may be unable to explain generalization in deep learning

**Vaishnavh Nagarajan**
Department of Computer Science
Carnegie Mellon University
Pittsburgh, PA
vaishnavh@cs.cmu.edu

**J. Zico Kolter**
Department of Computer Science
Carnegie Mellon University &
Bosch Center for Artificial Intelligence
Pittsburgh, PA
zkolter@cs.cmu.edu

## Abstract

Aimed at explaining the surprisingly good generalization behavior of overparam-
eterized deep networks, recent works have developed a variety of generalization
bounds for deep learning, all based on the fundamental learning-theoretic technique
of uniform convergence. While it is well-known that many of these existing bounds
are numerically large, through numerous experiments, we bring to light a more
concerning aspect of these bounds: in practice, these bounds can *increase* with
the training dataset size. Guided by our observations, we then present examples
of overparameterized linear classifiers and neural networks trained by gradient
descent (GD) where uniform convergence provably cannot "explain generalization"
– even if we take into account the implicit bias of GD *to the fullest extent possible*.
More precisely, even if we consider only the set of classifiers output by GD, which
have test errors less than some small $\epsilon$ in our settings, we show that applying
(two-sided) uniform convergence on this set of classifiers will yield only a vacuous
generalization guarantee larger than $1 - \epsilon$. Through these findings, we cast doubt
on the power of uniform convergence-based generalization bounds to provide a
complete picture of why overparameterized deep networks generalize well.

## 1 Introduction

Explaining why overparameterized deep networks generalize well [28, 38] has become an important
open question in deep learning. How is it possible that a large network can be trained to perfectly fit
randomly labeled data (essentially by memorizing the labels), and yet, the same network when trained
to perfectly fit real training data, generalizes well to unseen data? This called for a "rethinking" of
conventional, algorithm-*independent* techniques to explain generalization. Specifically, it was argued
that learning-theoretic approaches must be reformed by identifying and incorporating the implicit
bias/regularization of stochastic gradient descent (SGD) [6, 35, 30]. Subsequently, a huge variety of
novel and refined, algorithm-*dependent* generalization bounds for deep networks have been developed,
all based on *uniform convergence*, the most widely used tool in learning theory. The ultimate goal
of this ongoing endeavor is to derive bounds on the generalization error that (a) are small, ideally
non-vacuous (i.e., $< 1$), (b) reflect the same width/depth dependence as the generalization error (e.g.,
become smaller with increasing width, as has been surprisingly observed in practice), (c) apply to the
network learned by SGD (without any modification or explicit regularization) and (d) increase with
the proportion of randomly flipped training labels (i.e., increase with memorization).

While every bound meets some of these criteria (and sheds a valuable but partial insight into
generalization in deep learning), there is no known bound that meets all of them simultaneously.
While most bounds [29, 3, 11, 31, 27, 32] apply to the original network, they are neither numerically
small for realistic dataset sizes, nor exhibit the desired width/depth dependencies (in fact, these

bounds grow exponentially with the depth). The remaining bounds hold either only on a compressed network [2] or a stochastic network [21] or a network that has been further modified via optimization or more than one of the above [8, 39]. Extending these bounds to the original network is understood to be highly non-trivial [27]. While strong width-independent bounds have been derived for two-layer ReLU networks [23, 1], these rely on a carefully curated, small learning rate and/or large batch size. (We refer the reader to Appendix A for a tabular summary of these bounds.)

In our paper, we bring to light another fundamental issue with existing bounds. We demonstrate that these bounds violate another natural but largely overlooked criterion for explaining generalization: (e) the bounds should decrease with the dataset size at the same rate as the generalization error. In fact, we empirically observe that these bounds can *increase* with dataset size, which is arguably a more concerning observation than the fact that they are large for a specific dataset size.

Motivated by the seemingly insurmountable hurdles towards developing bounds satisfying all the above five necessary criteria, we take a step back and examine how the underlying technique of uniform convergence may itself be inherently limited in the overparameterized regime. Specifically, we present examples of overparameterized linear classifiers and neural networks trained by GD (or SGD) where uniform convergence can *provably* fail to explain generalization. Intuitively, our examples highlight that overparameterized models trained by gradient descent can learn decision boundaries that are largely "simple" – and hence generalize well – but have "microscopic complexities" which cannot be explained away by uniform convergence. Thus our results call into question the active ongoing pursuit of using uniform convergence to fully explain generalization in deep learning.

**Our contributions in more detail.** We first show that in practice certain weight norms of deep ReLU networks, such as the distance from initialization, *increase* polynomially with the number of training examples (denoted by $m$). We then show that as a result, existing generalization bounds – all of which depend on such weight norms – fail to reflect even a dependence on $m$ even reasonably similar to the actual test error, violating criterion (e); for sufficiently small batch sizes, these bounds even grow with the number of examples. This observation uncovers a conceptual gap in our understanding of the puzzle, by pointing towards a source of vacuity unrelated to parameter count.

As our second contribution, we consider three example setups of overparameterized models trained by (stochastic) gradient descent – a linear classifier, a sufficiently wide neural network with ReLUs and an infinite width neural network with exponential activations (with the hidden layer weights frozen) – that learn some underlying data distribution with small generalization error (say, at most $\epsilon$). These settings also simulate our observation that norms such as distance from initialization grow with dataset size $m$. More importantly, we prove that, in these settings, *any* two-sided uniform convergence bound would yield a (nearly) vacuous generalization bound.

Notably, this vacuity holds even if we "aggressively" take implicit regularization into account while applying uniform convergence – described more concretely as follows. Recall that roughly speaking a uniform convergence bound essentially evaluates the complexity of a hypothesis class (see Definition 3.2). As suggested by Zhang et al. [38], one can tighten uniform convergence bounds by pruning the hypothesis class to remove extraneous hypotheses never picked by the learning algorithm for the data distribution of interest. In our setups, even if we apply uniform convergence on the set of only those hypotheses picked by the learner whose test errors are all negligible (at most $\epsilon$), one can get no better than a nearly vacuous bound on the generalization error (that is at least $1 - \epsilon$). In this sense, we say that uniform convergence provably cannot explain generalization in our settings. Finally, we note that while nearly all existing uniform convergence-based techniques are two-sided, we show that even PAC-Bayesian bounds, which are typically presented only as one-sided convergence, also boil down to nearly vacuous guarantees in our settings.

## 1.1 Related Work

**Weight norms vs. training set size $m$.** Prior works like Neyshabur et al. [30] and Nagarajan and Kolter [26] have studied the behavior of weight norms in deep learning. Although these works do not explicitly study the dependence of these norms on training set size $m$, one can infer from their plots that weight norms of deep networks show some increase with $m$. Belkin et al. [4] reported a similar paradox in kernel learning, observing that norms that appear in kernel generalization bounds increase with $m$, and that this is due to noise in the labels. Kawaguchi et al. [19] showed that there exist linear models with arbitrarily large weight norms that can generalize well, although such weights are not

necessarily found by gradient descent. We crucially supplement these observations in three ways. First, we empirically and theoretically demonstrate how, even with *zero label noise* (unlike [4]) and by gradient descent (unlike [19]), a significant level of $m$-dependence can arise in the weight norms – significant enough to make even the generalization bound grow with $m$. Next, we identify uniform convergence as the root cause behind this issue, and thirdly and most importantly, we provably demonstrate this is so.

**Weaknesses of Uniform Convergence.** Traditional wisdom is that uniform convergence bounds are a bad choice for complex classifiers like k-nearest neighbors because these hypotheses classes have infinite VC-dimension (which motivated the need for stability based generalization bounds in these cases [33, 5]). However, this sort of an argument against uniform convergence may still leave one with the faint hope that, by aggressively pruning the hypothesis class (depending on the algorithm and the data distribution), one can achieve meaningful uniform convergence. In contrast, we seek to rigorously and thoroughly rule out uniform convergence in the settings we study. We do this by first defining the tightest form of uniform convergence in Definition 3.3 – one that lower bounds any uniform convergence bound – and then showing that even this bound is vacuous in our settings. Additionally, we note that we show this kind of failure of uniform convergence for linear classifiers, which is a much simpler model compared to k-nearest neighbors.

For deep networks, Zhang et al. [38] showed that applying uniform convergence on the whole hypothesis class fails, and that it should instead be applied in an algorithm-dependent way. *Ours is a much different claim* – that uniform convergence is inherently problematic in that even the algorithm-dependent application would fail – casting doubt on the rich line of post-Zhang et al. [38] algorithm-dependent approaches. At the same time, we must add the disclaimer that our results do not preclude the fact that uniform convergence may still work if GD is run with explicit regularization (such as weight decay). Such a regularized setting however, is not the main focus of the generalization puzzle [38, 28].

Prior works [36, 34] have also focused on understanding uniform convergence for *learnability of learning problems*. Roughly speaking, learnability is a strict notion that does not have to hold even though an algorithm may generalize well for simple distributions in a learning problem. While we defer the details of these works in Appendix I, we emphasize here that these results are orthogonal to (i.e., neither imply nor contradict) our results.

## 2 Existing bounds vs. training set size

As we stated in criterion (e) in the introduction, a fundamental requirement from a generalization bound, however numerically large the bound may be, is that it should vary inversely with the size of the training dataset size $(m)$ like the observed generalization error. Such a requirement is satisfied even by standard parameter-count-based VC-dimension bounds, like $\mathcal{O}(dh/\sqrt{m})$ for depth $d$, width $h$ ReLU networks [13]. Recent works have "tightened" the parameter-count-dependent terms in these bounds by replacing them with seemingly innocuous norm-based quantities; however, we show below that this has also inadvertently introduced training-set-size-count dependencies in the numerator, contributing to the vacuity of bounds. With these dependencies, the generalization bounds even increase with training dataset size for small batch sizes.

**Setup and notations.** We focus on fully connected networks of depth $d = 5$, width $h = 1024$ trained on MNIST, although we consider other settings in Appendix B. We use SGD with learning rate $0.1$ and batch size 1 to minimize cross-entropy loss until $99\%$ of the training data are classified correctly by a *margin* of at least $\gamma^\star = 10$ i.e., if we denote by $f(\mathbf{x})[y]$ the real-valued logit output (i.e., pre-softmax) on class $y$ for an input $\mathbf{x}$, we ensure that for $99\%$ of the data $(\mathbf{x}, y)$, the margin $\Gamma(f(\mathbf{x}), y) := f(\mathbf{x})[y] - \max_{y' \neq y} f(\mathbf{x})[y']$ is at least $\gamma^\star$. We emphasize that, from the perspective of generalization guarantees, this stopping criterion helps standardize training across different hyperparameter values, including different values of $m$ [30]. Now, observe that for this particular stopping criterion, the test error empirically decreases with size $m$ as $1/m^{0.43}$ as seen in Figure 1 (third plot). However, we will see that the story is starkly different for the generalization bounds.

**Norms grow with training set size $m$.** Before we examine the overall generalization bounds themselves, we first focus on two quantities that recur in the numerator of many recent bounds: the $\ell_2$ distance of the weights from their initialization [8, 26] and the product of spectral norms of the weight

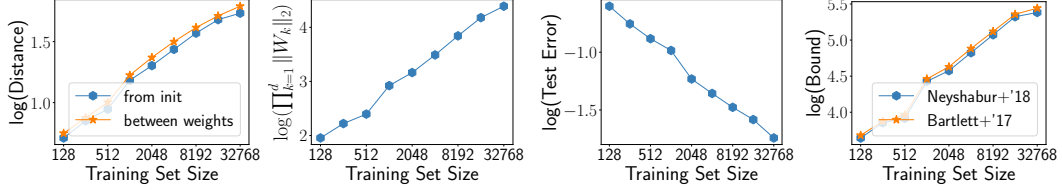

Figure 1: **Experiments in Section 2:** In the **first** figure, we plot (i) $\ell_2$ the distance of the network from the initialization and (ii) the $\ell_2$ distance between the weights learned on two random draws of training data starting from the same initialization. In the **second** figure we plot the product of spectral norms of the weights matrices. In the **third** figure, we plot the test error. In the **fourth** figure, we plot the bounds from [31, 3]. Note that we have presented log-log plots and the exponent of $m$ can be recovered from the slope of these plots.

matrices of the network [31, 3]. We observe in Figure 1 (first two plots, blue lines) that both these quantities grow at a polynomial rate with $m$: the former at the rate of at least $m^{0.4}$ and the latter at a rate of $m$. Our observation is a follow-up to Nagarajan and Kolter [26] who argued that while distance of the parameters from the origin grows with width as $\Omega(\sqrt{h})$, the distance from initialization is width-independent (and even decreases with width); hence, they concluded that incorporating the initialization would improve generalization bounds by a $\Omega(\sqrt{h})$ factor. However, our observations imply that, even though distance from initialization would help explain generalization better in terms of width, it conspicuously fails to help explain generalization in terms of its dependence on $m$ (and so does distance from origin as we show in Appendix Figure 5). [1]

Additionally, we also examine another quantity as an alternative to distance from initialization: the $\ell_2$ diameter of *the parameter space explored by SGD*. That is, for a fixed initialization and data distribution, we consider the set of all parameters learned by SGD across all draws of a dataset of size $m$; we then consider the diameter of the smallest ball enclosing this set. If this diameter exhibits a better behavior than the above quantities, one could then explain generalization better by replacing the distance from initialization with the distance from the center of this ball in existing bounds. As a lower bound on this diameter, we consider the distance between the weights learned on two independently drawn datasets from the given initialization. Unfortunately, we observe that even this quantity shows a similar undesirable behavior with respect to $m$ like distance from initialization (see Figure 1, first plot, orange line).

**The bounds grow with training set size $m$.** We now turn to evaluating existing guarantees from Neyshabur et al. [31] and Bartlett et al. [3]. As we note later, our observations apply to many other bounds too. Let $W_1, \ldots, W_d$ be the weights of the learned network (with $W_1$ being the weights adjacent to the inputs), $Z_1, \ldots, Z_d$ the random initialization, $\mathcal{D}$ the true data distribution and $S$ the training dataset. For all inputs $\mathbf{x}$, let $\|\mathbf{x}\|_2 \leq B$. Let $\|\cdot\|_2, \|\cdot\|_F, \|\cdot\|_{2,1}$ denote the spectral norm, the Frobenius norm and the matrix $(2, 1)$-norm respectively; let $\mathbf{1}[\cdot]$ be the indicator function. Recall that $\Gamma(f(\mathbf{x}), y) := f(\mathbf{x})[y] - \max_{y' \neq y} f(\mathbf{x})[y']$ denotes the margin of the network on a datapoint. Then, for any constant $\gamma$, these generalization guarantees are written as follows, ignoring log factors:

$$\Pr_{\mathcal{D}}[\Gamma(f(\mathbf{x}), y) \leq 0] \leq \frac{1}{m} \sum_{(x,y) \in S} \mathbf{1}[\Gamma(f(\mathbf{x}), y) \leq \gamma] + \text{generalization error bound}. \qquad (1)$$

Here the generalization error bound is of the form $\mathcal{O}\left(\frac{Bd\sqrt{h}}{\gamma\sqrt{m}} \prod_{k=1}^{d} \|W_k\|_2 \times \texttt{dist}\right)$ where $\texttt{dist}$ equals $\sqrt{\sum_{k=1}^{d} \frac{\|W_k - Z_k\|_F^2}{\|W_k\|_2^2}}$ in [31] and $\frac{1}{d\sqrt{h}} \left(\sum_{k=1}^{d} \left(\frac{\|W_k - Z_k\|_{2,1}}{\|W_k\|_2}\right)^{2/3}\right)^{3/2}$ in [3].

In our experiments, since we train the networks to fit at least $99\%$ of the datapoints with a margin of 10, in the above bounds, we set $\gamma = 10$ so that the first train error term in the right hand side of Equation 1 becomes a small value of at most 0.01. We then plot in Figure 1 (fourth plot), the second

term above, namely the generalization error bounds, and observe that all these bounds *grow with the sample size* $m$ as $\Omega(m^{0.68})$, thanks to the fact that the terms in the numerator of these bounds grow with $m$. Since we are free to plug in $\gamma$ in Equation 1, one may wonder whether there exists a better choice of $\gamma$ for which we can observe a smaller increase on $m$ (since the plotted terms inversely depend on $\gamma$). However, in Appendix Figure 6 we establish that even for larger values of $\gamma$, this $m$-dependence remains. Also note that, although we do not plot the bounds from [27, 11], these have nearly identical norms in their numerator, and so one would not expect these bounds to show radically better behavior with respect to $m$. Finally, we defer experiments conducted for other varied settings, and the neural network bound from [32] to Appendix B.

While the bounds might show better $m$-dependence for other settings – indeed, for larger batches, we show in Appendix B that the bounds behave better – we believe that the egregious break down of these bounds in this setting (and many other hyperparameter settings as presented in Appendix B) must imply fundamental issues with the bounds themselves. While this may be addressed to some extent with a better understanding of implicit regularization in deep learning, we regard our observations as a call for taking a step back and clearly understanding any inherent limitations in the theoretical tool underlying all these bounds namely, uniform convergence. [2]

## 3 Provable failure of uniform convergence

**Preliminaries.** Let $\mathcal{H}$ be a class of hypotheses mapping from $\mathcal{X}$ to $\mathbb{R}$, and let $\mathcal{D}$ be a distribution over $\mathcal{X} \times \{-1, +1\}$. The loss function we mainly care about is the 0-1 error; but since a direct analysis of the uniform convergence of the 0-1 error is hard, sometimes a more general margin-based surrogate of this error (also called as ramp loss) is analyzed for uniform convergence. Specifically, given the classifier's logit output $y' \in \mathbb{R}$ and the true label $y \in \{-1, +1\}$, define

$$\mathcal{L}^{(\gamma)}(y', y) = \begin{cases} 1 & yy' \leq 0 \\ 1 - \frac{yy'}{\gamma} & yy' \in (0, \gamma) \\ 0 & yy' \geq \gamma. \end{cases}$$

Note that $\mathcal{L}^{(0)}$ is the 0-1 error, and $\mathcal{L}^{(\gamma)}$ an upper bound on the 0-1 error. We define for any $\mathcal{L}$, the expected loss as $\mathcal{L}_{\mathcal{D}}(h) := \mathbb{E}_{(\mathbf{x}, y) \sim \mathcal{D}}[\mathcal{L}(h(\mathbf{x}), y)]$ and the empirical loss on a dataset $S$ of $m$ datapoints $\hat{\mathcal{L}}_S(h) := \frac{1}{m} \sum_{(\mathbf{x}, y) \in S} \mathcal{L}(h(\mathbf{x}), y)$. Let $\mathcal{A}$ be the learning algorithm and let $h_S$ be the hypothesis output by the algorithm on a dataset $S$ (assume that any training-data-independent randomness, such as the initialization/data-shuffling is fixed).

For a given $\delta \in (0, 1)$, the generalization error of the algorithm is essentially a bound on the difference between the error of the hypothesis $h_S$ learned on a training set $S$ and the expected error over $\mathcal{D}$, that holds with high probability of at least $1 - \delta$ over the draws of $S$. More formally:

**Definition 3.1.** The **generalization error** of $\mathcal{A}$ with respect to loss $\mathcal{L}$ is the smallest value $\epsilon_{\mathrm{gen}}(m, \delta)$ such that: $\mathrm{Pr}_{S \sim \mathcal{D}^m} \left[ \mathcal{L}_{\mathcal{D}}(h_S) - \hat{\mathcal{L}}_S(h_S) \leq \epsilon_{\mathrm{gen}}(m, \delta) \right] \geq 1 - \delta$.

To theoretically bound the generalization error of the algorithm, the most common approach is to provide a two-sided uniform convergence bound on the hypothesis class used by the algorithm, where, for a given draw of $S$, we look at convergence for all the hypotheses in $\mathcal{H}$ instead of just $h_S$:

**Definition 3.2.** The **uniform convergence bound** with respect to loss $\mathcal{L}$ is the smallest value $\epsilon_{\mathrm{unif}}(m, \delta)$ such that: $\mathrm{Pr}_{S \sim \mathcal{D}^m} \left[ \sup_{h \in \mathcal{H}} \left| \mathcal{L}_{\mathcal{D}}(h) - \hat{\mathcal{L}}_S(h) \right| \leq \epsilon_{\mathrm{unif}}(m, \delta) \right] \geq 1 - \delta$.

**Tightest algorithm-dependent uniform convergence.** The bound given by $\epsilon_{\text{unif}}$ can be tightened by ignoring many extraneous hypotheses in $\mathcal{H}$ never picked by $\mathcal{A}$ for a given simple distribution $\mathcal{D}$. This is typically done by focusing on a norm-bounded class of hypotheses that the algorithm $\mathcal{A}$ implicitly restricts itself to. Let us take this to the extreme by applying uniform convergence on "the smallest possible class" of hypotheses, namely, *only* those hypotheses that are picked by $\mathcal{A}$ under $\mathcal{D}$, excluding everything else. Observe that pruning the hypothesis class any further would not imply a bound on the generalization error, and hence applying uniform convergence on this aggressively pruned hypothesis class would yield the tightest possible uniform convergence bound. Recall that we care about this formulation because our goal is to rigorously and thoroughly rule out the possibility that no kind of uniform convergence bound, however cleverly applied, can explain generalization in our settings of interest (which we will describe later).

To formally capture this bound, it is helpful to first rephrase the above definition of $\epsilon_{\text{unif}}$: we can say that $\epsilon_{\text{unif}}(m, \delta)$ is the smallest value for which there exists a set of sample sets $\mathcal{S}_\delta \subseteq (\mathcal{X} \times \{-1, 1\})^m$ for which $Pr_{S \sim \mathcal{D}^m}[S \in \mathcal{S}_\delta] \geq 1 - \delta$ and furthermore, $\sup_{S \in \mathcal{S}_\delta} \sup_{h \in \mathcal{H}} |\mathcal{L}_\mathcal{D}(h) - \hat{\mathcal{L}}_S(h)| \leq \epsilon_{\text{unif}}(m, \delta)$. Observe that this definition is *equivalent* to Definition 3.2. Extending this rephrased definition, we can define the tightest uniform convergence bound by replacing $\mathcal{H}$ here with only those hypotheses that are explored by the algorithm $\mathcal{A}$ under the datasets belonging to $\mathcal{S}_\delta$:

**Definition 3.3.** The **tightest algorithm-dependent uniform convergence bound** with respect to loss $\mathcal{L}$ is the smallest value $\epsilon_{\text{unif-alg}}(m, \delta)$ for which there exists a set of sample sets $\mathcal{S}_\delta$ such that $Pr_{S \sim \mathcal{D}^m}[S \in \mathcal{S}_\delta] \geq 1 - \delta$ and if we define the space of hypotheses explored by $\mathcal{A}$ on $\mathcal{S}_\delta$ as $\mathcal{H}_\delta := \bigcup_{S \in \mathcal{S}_\delta}\{h_S\} \subseteq \mathcal{H}$, the following holds: $\sup_{S \in \mathcal{S}_\delta} \sup_{h \in \mathcal{H}_\delta} \left|\mathcal{L}_\mathcal{D}(h) - \hat{\mathcal{L}}_S(h)\right| \leq \epsilon_{\text{unif-alg}}(m, \delta)$.

In the following sections, through examples of overparameterized models trained by GD (or SGD), we argue how even the above tightest algorithm-dependent uniform convergence can fail to explain generalization. i.e., in these settings, even though $\epsilon_{\text{gen}}$ is smaller than a negligible value $\epsilon$, we show that $\epsilon_{\text{unif-alg}}$ is large (specifically, at least $1 - \epsilon$). Before we delve into these examples, below we quickly outline the key mathematical idea by which uniform convergence is made to fail.

Consider a scenario where the algorithm generalizes well i.e., for every training set $\tilde{S}$, $h_{\tilde{S}}$ has zero error on $\tilde{S}$ and has small test error. While this means that $h_{\tilde{S}}$ has small error on *random* draws of a test set, it may still be possible that for every such $h_{\tilde{S}}$, there exists a corresponding "bad" dataset $\tilde{S}'$ – that is not random, but rather dependent on $\tilde{S}$ – on which $h_{\tilde{S}}$ has a large empirical error (say 1). Unfortunately, uniform convergence runs into trouble while dealing with such bad datasets. Specifically, as we can see from the above definition, uniform convergence demands that $|\mathcal{L}_\mathcal{D}(h_{\tilde{S}}) - \hat{\mathcal{L}}_S(h_{\tilde{S}})|$ be small on all datasets in $\mathcal{S}_\delta$, which excludes a $\delta$ fraction of the datasets. While it may be tempting to think that we can somehow exclude the bad dataset as part of the $\delta$ fraction, there is a significant catch here: we can not carve out a $\delta$ fraction specific to each hypothesis; we can ignore only a single chunk of $\delta$ mass common to all hypotheses in $\mathcal{H}_\delta$. This restriction turns out to be a tremendous bottleneck: despite ignoring this $\delta$ fraction, for most $h_{\tilde{S}} \in \mathcal{H}_\delta$, the corresponding bad set $\tilde{S}'$ would still be left in $\mathcal{S}_\delta$. Then, for all such $h_{\tilde{S}}$, $\mathcal{L}_\mathcal{D}(h_{\tilde{S}})$ would be small but $\hat{\mathcal{L}}_S(h_{\tilde{S}})$ large; we can then set the $S$ inside the $\sup_{S \in \mathcal{S}_\delta}$ to be $\tilde{S}'$ to conclude that $\epsilon_{\text{unif-alg}}$ is indeed vacuous. This is the kind of failure we will demonstrate in a high-dimensional linear classifier in the following section, and a ReLU neural network in Section 3.2, and an infinitely wide exponential-activation neural network in Appendix F – all trained by GD or SGD. [3]

**Note:** Our results about failure of uniform convergence holds even for bounds that output a different value for each hypothesis. In this case, the tightest uniform convergence bound for a given hypothesis would be at least as large as $\sup_{S \in \mathcal{S}_\delta} |\mathcal{L}_\mathcal{D}(h_{\tilde{S}}) - \hat{\mathcal{L}}_S(h_{\tilde{S}})|$ which by a similar argument would be vacuous for most draws of the training set $\tilde{S}$. We discuss this in more detail in Appendix G.4.

## 3.1 High-dimensional linear classifier

**Why a linear model?** Although we present a neural network example in the next section, we first emphasize why it is also important to understand how uniform convergence could fail for linear

classifiers trained using GD. First, it is more natural to expect uniform convergence to yield poorer bounds in more complicated classifiers; linear models are arguably the simplest of classifiers, and hence showing failure of uniform convergence in these models is, in a sense, the most interesting. Secondly, recent works (e.g., [17]) have shown that as the width of a deep network goes to infinity, under some conditions, the network converges to a high-dimensional linear model (trained on a high-dimensional transformation of the data) – thus making the study of high-dimensional linear models relevant to us. Note that our example is not aimed at modeling the setup of such linearized neural networks. However, it does provide valuable intuition about the mechanism by which uniform convergence fails, and we show how this extends to neural networks in the later sections.

**Setup.** Let each input be a $K + D$ dimensional vector (think of $K$ as a small constant and $D$ much larger than $m$). The value of any input $\mathbf{x}$ is denoted by $(\mathbf{x}_1, \mathbf{x}_2)$ where $\mathbf{x}_1 \in \mathbb{R}^K$ and $\mathbf{x}_2 \in \mathbb{R}^D$. Let the centers of the (two) classes be determined by an arbitrary vector $\mathbf{u} \in \mathbb{R}^K$ such that $\|\mathbf{u}\|_2 = 1/\sqrt{m}$. Let $\mathcal{D}$ be such that the label $y$ has equal probability of being $+1$ and $-1$, and $\mathbf{x}_1 = 2 \cdot y \cdot \mathbf{u}$ while $\mathbf{x}_2$ is sampled independently from a spherical Gaussian, $\mathcal{N}(0, \frac{32}{D}I)$.[4] Note that the distribution is linearly separable based on the first few ($K$) dimensions. For the learning algorithm $\mathcal{A}$, consider a linear classifier with weights $\mathbf{w} = (\mathbf{w}_1, \mathbf{w}_2)$ and whose output is $h(\mathbf{x}) = \mathbf{w}_1\mathbf{x}_1 + \mathbf{w}_2\mathbf{x}_2$. Assume the weights are initialized to the origin. Given a dataset $S$, $\mathcal{A}$ takes a gradient step of learning rate 1 to maximize $y \cdot h(\mathbf{x})$ for each $(\mathbf{x}, y) \in S$. Hence, regardless of the batch size, the learned weights would satisfy, $\mathbf{w}_1 = 2m\mathbf{u}$ and $\mathbf{w}_2 = \sum_i y^{(i)}\mathbf{x}_2^{(i)}$. Note that effectively $\mathbf{w}_1$ is aligned correctly along the class boundary while $\mathbf{w}_2$ is high-dimensional Gaussian noise. It is fairly simple to show that this algorithm achieves *zero training error* for most draws of the training set. At the same time, for this setup, we have the following lower bound on uniform convergence for the $\mathcal{L}^{(\gamma)}$ loss:[5]

**Theorem 3.1.** *For any $\epsilon, \delta > 0, \delta \leq 1/4$, when $D = \Omega\left(\max\left(m \ln \frac{m}{\delta}, m \ln \frac{1}{\epsilon}\right)\right)$, $\gamma \in [0,1]$, the $\mathcal{L}^{(\gamma)}$ loss satisfies $\epsilon_{gen}(m, \delta) \leq \epsilon$, while $\epsilon_{unif\text{-}alg}(m, \delta) \geq 1 - \epsilon$. Furthermore, for all $\gamma \geq 0$, for the $\mathcal{L}^{(\gamma)}$ loss, $\epsilon_{unif\text{-}alg}(m, \delta) \geq 1 - \epsilon_{gen}(m, \delta)$.*

In other words, even the tightest uniform convergence bound is nearly vacuous despite good generalization. In order to better appreciate the implications of this statement, it will be helpful to look at the bound a standard technique would yield here. For example, the Rademacher complexity of the class of $\ell_2$-norm bounded linear classifiers would yield a bound of the form $\mathcal{O}(\|\mathbf{w}\|_2/(\gamma^\star \sqrt{m}))$ where $\gamma^\star$ is the margin on the training data. In this setup, the weight norm grows with dataset size as $\|\mathbf{w}\|_2 = \Theta(\sqrt{m})$ (which follows from the fact that $\mathbf{w}_2$ is a Gaussian with $m/D$ variance along each of the $D$ dimensions) and $\gamma^\star = \Theta(1)$. Hence, the Rademacher bound here would evaluate to a constant much larger than $\epsilon$. One might persist and think that perhaps, the characterization of $\mathbf{w}$ to be bounded in $\ell_2$ norm does not fully capture the implicit bias of the algorithm. Are there other properties of the Gaussian $\mathbf{w}_2$ that one could take into account to identify an even smaller class of hypotheses for which uniform convergence may work after all? Unfortunately, our statement rules this out: even after fixing $\mathbf{w}_1$ to the learned value ($2m\mathbf{u}$) and for any possible $1 - \delta$ truncation of the Gaussian $\mathbf{w}_2$, the resulting pruned class of weights – despite all of them having a test error less than $\epsilon$ – would give only nearly vacuous uniform convergence bounds as $\epsilon_{unif\text{-}alg}(m, \delta) \geq 1 - \epsilon$.

**Proof outline.** We now provide an outline of our argument for Theorem 3.1, deferring the proof to the appendix. First, the small generalization (and test) error arises from the fact that $\mathbf{w}_1$ is aligned correctly along the true boundary; at the same time, the noisy part of the classifier $\mathbf{w}_2$ is poorly aligned with at least $1 - \epsilon$ mass of the test inputs, and hence does not dominate the output of the classifier on test data – preserving the good fit of $\mathbf{w}_1$ on the test data. On the other hand, at a very high level, under the purview of uniform convergence, we can argue that the noise vector $\mathbf{w}_2$ is effectively stripped of its randomness. This misleads uniform convergence into believing that the $D$ noisy dimensions (where $D > m$) contribute meaningfully to the representational complexity of the classifier, thereby giving nearly vacuous bounds. We describe this more concretely below.

As a key step in our argument, we show that w.h.p over draws of $S$, even though the learned classifier $h_S$ correctly classifies most of the randomly picked test data, it completely misclassifies

a "bad" dataset, namely $S' = \{((\mathbf{x}_1, -\mathbf{x}_2), y) \mid (\mathbf{x}, y) \in S\}$ which is the noise-negated version of $S$. Now recall that to compute $\epsilon_{\text{unif-alg}}$ one has to begin by picking a sample set space $\mathcal{S}_\delta$ of mass $1 - \delta$. We first argue that for *any* choice of $\mathcal{S}_\delta$, there must exist $S_\star$ such that all the following four events hold: (i) $S_\star \in \mathcal{S}_\delta$, (ii) the noise-negated $S'_\star \in \mathcal{S}_\delta$, (iii) $h_{S_\star}$ has test error less than $\epsilon$ and (iv) $h_{S_\star}$ completely misclassifies $S'_\star$. We prove the existence of such an $S_\star$ by arguing that over draws from $\mathcal{D}^m$, there is non-zero probability of picking a dataset that satisfies these four conditions. Note that our argument for this crucially makes use of the fact that we have designed the "bad" dataset in a way that it has the same distribution as the training set, namely $\mathcal{D}^m$. Finally, for a given $\mathcal{S}_\delta$, if we have an $S_\star$ satisfying (i) to (iv), we can prove our claim as $\epsilon_{\text{unif-alg}}(m, \delta) = \sup_{S \in \mathcal{S}_\delta} \sup_{h \in \mathcal{H}_\delta} |\mathcal{L}_\mathcal{D}(h) - \hat{\mathcal{L}}_S(h)| \geq |\mathcal{L}_\mathcal{D}(h_{S_\star}) - \hat{\mathcal{L}}_{S'_\star}(h_{S_\star})| = |\epsilon - 1| = 1 - \epsilon$.

**Remark 3.1.** Our analysis depends on the fact that $\epsilon_{\text{unif-alg}}$ is a two-sided convergence bound – which is what existing techniques bound – and our result would not apply for hypothetical one-sided uniform convergence bounds. While PAC-Bayes based bounds are typically presented as one-sided bounds, we show in Appendix J that even these are lower-bounded by the two-sided $\epsilon_{\text{unif-alg}}$. To the best of our knowledge, it is non-trivial to make any of these tools purely one-sided.

**Remark 3.2.** The classifier modified by setting $\mathbf{w}_2 \leftarrow 0$, has small test error and also enjoys non-vacuous bounds as it has very few parameters. However, such a bound would not fully explain why the original classifier generalizes well. One might then wonder if such a bound could be extended to the original classifier, like it was explored in Nagarajan and Kolter [27] for deep networks. Our result implies that no such extension is possible in this particular example.

## 3.2 ReLU neural network

We now design a non-linearly separable task (with no "noisy" dimensions) where a sufficiently wide ReLU network trained in the standard manner, like in the experiments of Section 2 leads to failure of uniform convergence. For our argument, we will rely on a classifier trained *empirically*, in contrast to our linear examples where we rely on an analytically derived expression for the learned classifier. Thus, this section illustrates that the effects we modeled theoretically in the linear classifier *are* indeed reflected in typical training settings, even though here it is difficult to precisely analyze the learning process. We also refer the reader to Appendix F, where we present an example of a neural network with exponential activation functions for which we do derive a closed form expression.

**Setup.** We consider a 1000-dimensional data, where two classes are distributed uniformly over two origin-centered hyperspheres with radius 1 and 1.1 respectively. We vary the number of training examples from $4k$ to $65k$ (thus ranging through typical dataset sizes like that of MNIST). Observe that compared to the linear example, this data distribution is more realistic in two ways. First, we do not have specific dimensions in the data that are noisy and second, the data dimensionality here as such is a constant less than $m$. Given samples from this distribution, we train a two-layer ReLU network with $h = 100k$ to minimize cross entropy loss using SGD with learning rate 0.1 and batch size 64. We train the network until 99% of the data is classified by a margin of 10.

As shown in Figure 2 (blue line), in this setup, the 0-1 error (i.e., $\mathcal{L}^{(0)}$) as approximated by the test set, decreases with $m \in [2^{12}, 2^{16}]$ at the rate of $O(m^{-0.5})$. Now, to prove failure of uniform convergence, we empirically show that a completely misclassified "bad" dataset $S'$ can be constructed in a manner similar to that of the previous example. In this setting, we pick $S'$ by simply projecting every training datapoint on the inner hypersphere onto the outer and vice versa, and then flipping the labels. Then, as shown in Figure 2 (orange line), $S'$ is completely misclassified by the learned network. Furthermore, like in the previous example, we have $S' \sim \mathcal{D}^m$ because the distributions are uniform over the hyperspheres. Having established these facts, the rest of the argument follows like in the previous setting, implying failure of uniform convergence as in Theorem 3.1 here too.

In Figure 2 (right), we visualize how the learned boundaries are skewed around the training data in a way that $S'$ is misclassified. Note that $S'$ is misclassified even when it has as many as $60k$ points, and even though the network was not explicitly trained to misclassify those points. Intuitively, this demonstrates that the boundary learned by the ReLU network has sufficient complexity that hurts uniform convergence while not affecting the generalization error, at least in this setting. We discuss the applicability of this observation to other hyperparameter settings in Appendix G.2.

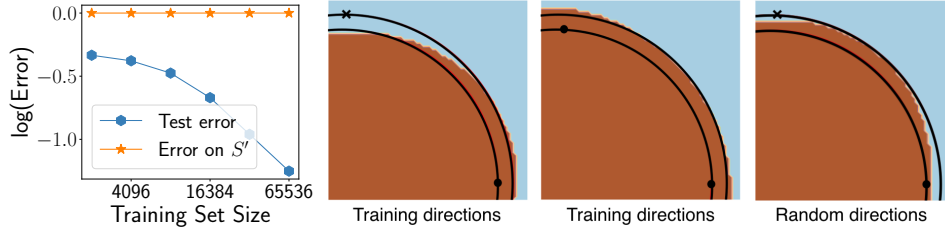

Figure 2: In the **first** figure, we plot the error of the ReLU network on test data and on the bad dataset $S'$, in the task described in Section 3.2. The **second and third** images correspond to the decision boundary learned in this task, in the 2D quadrant containing two training datapoints (depicted as $\times$ and $\bullet$). The black lines correspond to the two hyperspheres, while the brown and blue regions correspond to the class output by the classifier. Here, we observe that the boundaries are skewed around the training data in a way that it misclassifies the nearest point from the opposite class (corresponding to $S'$, that is not explicitly marked). The **fourth** image corresponds to two random (test) datapoints, where the boundaries are fairly random, and very likely to be located in between the hyperspheres (better confirmed by the low test error).

**Deep learning conjecture.**   Extending the above insights more generally, we conjecture that in overparameterized deep networks, SGD finds a fit that is simple at a macroscopic level (leading to good generalization) but also has many microscopic fluctuations (hurting uniform convergence). To make this more concrete, for illustration, consider the high-dimensional linear model that sufficiently wide networks have been shown to converge to [17]. That is, roughly, these networks can be written as $h(\mathbf{x}) = \mathbf{w}^T \phi(\mathbf{x})$ where $\phi(\mathbf{x})$ is a rich high-dimensional representation of $\mathbf{x}$ computed from many random features (chosen independent of training data). Inspired by our linear model in Section 3.1, we conjecture that the weights $\mathbf{w}$ learned on a dataset $S$ can be expressed as $\mathbf{w}_1 + \mathbf{w}_2$, where $\mathbf{w}_1^T \phi(\mathbf{x})$ dominates the output on most test inputs and induces a simple decision boundary. That is, it may be possible to apply uniform convergence on the function $\mathbf{w}_1^T \phi(\mathbf{x})$ to obtain a small generalization bound. On the other hand, $\mathbf{w}_2$ corresponds to meaningless signals that gradient descent gathered from the high-dimensional representation of the training set $S$. Crucially, these signals would be specific to $S$, and hence not likely to correlate with most of the test data i.e., $\mathbf{w}_2 \phi(\mathbf{x})$ would be negligible on most test data, thereby not affecting the generalization error significantly. However, $\mathbf{w}_2 \phi(\mathbf{x})$ can still create complex fluctuations on the boundary, in low-probability regions of the input space (whose locations would depend on $S$, like in our examples). As we argued, this can lead to failure of uniform convergence. Perhaps, existing works that have achieved strong uniform convergence bounds on modified networks, may have done so by implicitly suppressing $\mathbf{w}_2$, either by compression, optimization or stochasticization. Revisiting these works may help verify our conjecture.

## 4   Conclusion and Future Work

A growing variety of uniform convergence based bounds [29, 3, 11, 2, 31, 8, 39, 23, 1, 27, 32] have sought to explain generalization in deep learning. While these may provide partial intuition about the puzzle, we ask a critical, high level question: by pursuing this broad direction, is it possible to achieve the grand goal of a small generalization bound that shows appropriate dependence on the sample size, width, depth, label noise, and batch size? We cast doubt on this by first, empirically showing that existing bounds can surprisingly increase with training set size for small batch sizes. We then presented example setups, including that of a ReLU neural network, for which uniform convergence provably fails to explain generalization, even after taking implicit bias into account.

Future work in understanding implicit regularization in deep learning may be better guided with our knowledge of the sample-size-dependence in the weight norms. To understand generalization, it may also be promising to explore other learning-theoretic techniques like, say, algorithmic stability [9, 12, 5, 34] ; our linear setup might also inspire new tools. Overall, through our work, we call for going beyond uniform convergence to fully explain generalization in deep learning.

**Acknowledgements.**   Vaishnavh Nagarajan is supported by a grant from the Bosch Center for AI.

## Footnotes

[1]It may be tempting to think that our observations are peculiar to the cross-entropy loss for which the optimization algorithm diverges. But we observe that even for the squared error loss (Appendix B) where the optimization procedure does not diverge to infinity, distance from initialization grows with $m$.

[2]**Side note:** Before we proceed to the next section, where we blame uniform convergence for the above problems, we briefly note that we considered another simpler possibility. Specifically, we hypothesized that, for *some* (not all) existing bounds, the above problems could arise from an issue that does not involve uniform convergence, which we term as *pseudo-overfitting*. Roughly speaking, a classifier pseudo-overfits when its decision boundary is simple but its *real-valued* output has large "bumps" around some or all of its training datapoint. As discussed in Appendix C, deep networks pseudo-overfit only to a limited extent, and hence psuedo-overfitting does not provide a complete explanation for the issues faced by these bounds.

[3]In Appendix H, the reader can find a more abstract setting illustrating this mathematical idea more clearly.

[4]As noted in Appendix G.3, it is easy to extend the discussion by assuming that $\mathbf{x}_1$ is spread out around $2y\mathbf{u}$.

[5]While it is obvious from Theorem 3.1 that the bound is nearly vacuous for any $\gamma \in [0, 1]$, in Appendix G.1, we argue that even for any $\gamma \geq 1$, the guarantee is nearly vacuous, although in a slightly different sense.

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
