[Supplementary Material]

# A  Summary of existing generalization bounds.

In this section, we provide an informal summary of the properties of (some of the) existing generalization bounds for ReLU networks in Table 1.

| Bound | Norm dependencies | Parameter-count dependencies | Numerical value | Holds on original network? |
|---|---|---|---|---|
| Harvey et al. [13] | - | depth × width | Large | Yes |
| Bartlett et al. [3] Neyshabur et al. [31] | Product of spectral norms dist. from init. (not necessarily $\ell_2$) | poly(width) exp(depth) | Large | Yes |
| Neyshabur et al. [29] Golowich et al. [11] | Product of Frobenius norms $\ell_2$ dist. from init. | $\sqrt{\text{width}}^{\text{depth}}$ | Very large | Yes |
| Nagarajan and Kolter [27] | Jacobian norms $\ell_2$ dist. from init. Inverse pre-activations | poly(width) poly(depth) | Inverse pre-activations can be very large | Yes |
| Neyshabur et al. [32] for two-layer networks | Spectral norm (1st layer) $\ell_2$ Dist. from init (1st layer) Frobenius norm (2nd layer) | $\sqrt{\text{width}}$ | Small | Yes |
| Arora et al. [2] | Jacobian norms dist. from init. | poly(width) poly(depth) | Small | No. Holds on compressed network |
| Dziugaite and Roy [8] | dist. from init. Noise-resilience of network | - | Non-vacuous on MNIST | No. Holds on an optimized, stochastic network |
| Zhou et al. [39] | Heuristic compressibility & noise-resilience of network | - | Non-vacuous on ImageNet | No. Holds on an optimized, stochastic, heuristically compressed, network |
| Allen-Zhu et al. [1] | $L_{2,4}$ norm (1st layer) Frobenius norm (2nd layer) | - | Small for carefully scaled init. and learning rate | Yes |
| Li and Liang [23] | - | - | Small for carefully scaled batch size and learning rate | Yes |

Table 1: Summary of generalization bounds for ReLU networks. We note that the analysis in Li and Liang [23] relies on a sufficiently small learning rate ($\approx \mathcal{O}(1/m^{1.2})$) and large batch size ($\approx \Omega(\sqrt{m})$). Hence, the resulting bound cannot describe how generalization varies with any other hyperparameter, like training set size or width, with everything else fixed. A similar analysis in Allen-Zhu et al. [1] requires fixing the learning rate to be inversely proportional to width. Their bound decreases only as $\Omega(1/m^{0.16})$, although, the actual generalization error is typically as small as $\mathcal{O}(1/m^{0.43})$.

# B  More Experiments

In this section, we present more experiments along the lines of what we presented in Section 2.

**Layerwise dependence on $m$.**  Recall that in the main paper, we show how the distance from initialization and the product of spectral norms vary with $m$ for network with six layers. In Figure 3, we show how the terms grow with sample size $m$ for each layer individually. Our main observation is that the first layer suffers from the largest dependence on $m$.

**Distance between trajectories of shuffled datasets grows with $m$.**  In the main paper, we saw that the distance between the solutions learned on different draws of the dataset grow substantially with $m$. In Figure 5 (left), we show that even the distance between the solutions learned on the same draw, but a different shuffling of the dataset grows substantially with $m$.

**Flat minima**  We also relate our observations regarding distance between two independently learned weights to the popular idea of "flat minima". Interestingly, Figure 4 demonstrates that walking linearly from the weights learned on one dataset draw to that on another draw (from the same initialization) preserves the test error. Note that although a similar observation was made in Dräxler

Figure 3: We plot the distance from initialization and the spectral norm of each individual layer, and observe that the lowermost layer shows the greatest dependence on $m$.

Figure 4: We plot the test errors of the networks that lie on the straight line between two weights learned on two independent random draws of training data starting from the same initialization. We observe that all these intermediate networks have the same test error as the original networks themselves.

et al. [7], Garipov et al. [10], they show the existence of *non-linear* paths of good solutions between parameters learned from *different* initializations. Our observation on the other hand implies that for a fixed initialization, SGD explores the *same* basin in the test loss minimum across different training sets. As discussed in the main paper, this explored basin/space has larger $\ell_2$-width for larger $m$ giving rise to a "paradox": on one hand, wider minima are believed to result in, or at least correlate with better generalization [15, 14, 20], but on the other, a larger $\ell_2$-width of the explored space results in larger uniform convergence bounds, making it harder to explain generalization.

We note a similar kind of paradox concerning noise in training. Specifically, it is intriguing that on one hand, generalization is aided by larger learning rates and smaller batch sizes [18, 16, 20] due to increased noise in SGD. On the other, theoretical analyses benefit from the opposite; Allen-Zhu et al. [1] even explicitly regularize SGD for their three-layer-network result to help "forget false information" gathered by SGD. In other words, it seems that noise aids generalization, yet hinders attempts at explaining generalization. The intuition from our examples (such as the linear example) is that such "false information" could provably impair uniform convergence without affecting generalization.

**Frobenius norms grow with $m$ when $m \gg h$.** Some bounds like [11] depend on the Frobenius norms of the weight matrices (or the distance from origin), which as noted in [26] are in fact width-dependent, and grow as $\Omega(\sqrt{h})$. However, even these terms do grow with the number of samples in the regime where $m$ is larger than $h$. In Figure 5, we report the total distance from origin of the learned parameters for a network with $h = 256$ (we choose a smaller width to better emphasize the growth of this term with $m$); here, we see that for $m > 8192$, the distance from origin grows at a rate of $\Omega(m^{0.42})$ that is quite similar to what we observed for distance from initialization.

**Even a relaxed notion of margin does not address the $m$-dependency.** Recall that in the main paper, we computed the generalization error bound in Equation 1 by setting $\gamma$ to be $\gamma^\star$, the margin achieved by the network on at least 99% of the data. One may hope that by choosing a larger value of $\gamma$, this bound would become smaller, and that the $m$-dependence may improve. We consider this possibility by computing the median margin of the network over the training set (instead of

Figure 5: On the **left**, we plot the distance between the weights learned on the two different shuffles of the same dataset, and it grows as fast as the distance from initialization. On the **right**, we plot the distance of the weights from the origin, learned for a network of width $h = 256$ and depth $d = 6$; for sufficiently large $m$, this grows as $\Omega(m^{0.42})$.

Figure 6: In the **left** plot, we plot the bounds after setting $\gamma$ to be the median margin on the training data – these bounds grow as $\Omega(m^{0.48})$. In the **right** plot the median value of the margin $\Gamma(f(\mathbf{x}), y)$ on the training dataset and observe that it grows as $\mathcal{O}(m^{0.2})$.

the 1%-percentile'th margin) and substituting this in the second term in the right hand side of the guarantee in Equation 1. By doing this, the first margin-based train error term in the right hand side of Equation 1 would simplify to $0.5$ (as half the training data are misclassified by this large margin). Thereby we already forgo an explanation of half of the generalization behavior. At least we could hope that the second term no longer grows with $m$. Unfortunately, we observe in Figure 6 (left) that the bounds still grow with $m$. This is because, as shown in Figure 6 (right), the median margin value does not grow as fast with $m$ as the numerators of these bounds grow.

**Effect of depth.** We observed that as the network gets shallower the bounds show better dependence with $m$. As an extreme case, we consider a network with only one hidden layer, and with $h = 50000$. Here we also present a third bound, namely that of Neyshabur et al. [32], besides the two bounds discussed in the main paper. Specifically, if $Z_1, Z_2$ are the random initializations of the weight matrices in the network, the generalization error bound (the last term in Equation 1) here is of the following form, ignoring log factors:

$$\frac{\|W_2\|_F (\|W_1 - Z_1\|_F + \|Z_1\|_2)}{\gamma \sqrt{m}} + \frac{\sqrt{h}}{\sqrt{m}}.$$

The first term here is meant to be width-independent, while the second term clearly depends on the width and does decrease with $m$ at the rate of $m^{-0.5}$. Hence, in our plots in Figure 7, we only focus on the first term. We see that these bounds are almost constant and decrease at a minute rate of $\Omega(m^{-0.066})$ while the test errors decrease much faster, at the rate of $\mathcal{O}(m^{-0.35})$.

**Effect of width.** In Figure 8, we demonstrate that our observation that the bounds increase with $m$ extends to widths $h = 128$ and $h = 2000$ too.

Figure 7: On the **left**, we plot how the bounds vary with sample size for a single hidden layer network with $50k$ hidden units. We observe that these bounds are almost constant, and at best decrease at a meagre rate of $\Omega(m^{-0.066})$. On the **right**, we plot the test errors for this network and observe that it decreases with $m$ at the rate of at least $\mathcal{O}(m^{0.35})$.

Figure 8: On the **left**, we plot the bounds for varying $m$ for $h = 128$. All these bounds grow with $m$ as $\Omega(m^{0.94})$. On the **right**, we show a similar plot for $h = 2000$ and observe that the bounds grow as $\Omega(m^{0.79})$.

## B.1 Effect of batch size

**Bounds vs. batch size for fixed** $m$**.** In Figure 9, we show how the bounds vary with the batch size for a fixed sample size of $16384$. It turns out that even though the test error decreases with decreasing batch size (for our fixed stopping criterion), all these bounds *increase* (by a couple of orders of magnitude) with decreasing batch size. Again, this is because the terms like distance from initialization *increase* for smaller batch sizes (perhaps because of greater levels of noise in the updates). Overall, existing bounds do not reflect the same behavior as the actual generalization error in terms of their dependence on the batch size.

Figure 9: On the **left**, we plot the bounds for varying batch sizes for $m = 16384$ and observe that these bounds *decrease* by around $2$ orders of magnitude. On the **right**, we plot the test errors for varying batch sizes and observe that test error increases with batch size albeit slightly.

**Bounds vs.** $m$ **for batch size of** $32$**.** In the main paper, we only dealt with a small batch size of $1$. In Figure 10, we show bounds vs. sample size plots for a batch size of $32$. We observe that in this case, the bounds do decrease with sample size, although only at a rate of $\mathcal{O}(m^{-0.23})$ which is not as fast as the observed decrease in test error which is $\Omega(m^{-0.44})$. Our intuition as to why the bounds

behave better (in terms of $m$-dependence) in the larger batch size regime is that here the amount of noise in the parameter updates is much less compared to smaller batch sizes (and as we discussed earlier, uniform convergence finds it challenging to explain away such noise).

Figure 10: On the **left**, we plot the bounds for varying $m$ for a batch size of 32 and observe that these bounds do decrease with $m$ as $\mathcal{O}(1/m^{0.23})$. On the **right**, we plot the test errors for various $m$ for batch size 32 and observe that test error varies as $\Omega(1/m^{0.44})$.

**Squared error loss.** All the experiments presented so far deal with the cross-entropy loss, for which the optimization procedure ideally diverges to infinity; thus, one might suspect that our results are sensitive to the stopping criterion. It would therefore be useful to consider the squared error loss where the optimum on the training loss can be found in a finite distance away from the random initialization. Specifically, we consider the case where the squared error loss between the outputs of the network and the one-hot encoding of the true labels is minimized to a value of $0.05$ on average over the training data.

We observe in Figure 11 that even for this case, the distance from initialization and the spectral norms grow with the sample size at a rate of at least $m^{0.3}$. On the other hand, the test error decreases with sample size as $1/m^{0.38}$, indicating that even for the squared error loss, these terms hurt would hurt the generalization bound with respect to its dependence on $m$.

Figure 11: On the **left** we plot the distance from initialization and the distance between weights learned on two different random draws of the datasets, as a function of varying training set size $m$, when trained on the squared error loss. Both these quantities grow as $\Omega(m^{0.35})$. In the **middle**, we show how the product of spectral norms grow as $\Omega(m^{0.315})$ for sufficiently large $m \geq 2048$. On the **right**, we observe that the test error (i.e., the averaged squared error loss on the test data) decreases with $m$ as $\mathcal{O}(m^{-0.38})$.

## C Pseudo-overfitting

Recall that in the main paper, we briefly discussed a new notion that we call as pseudo-overfitting and noted that it is not the reason behind why some techniques lead to vacuous generalization bounds. We describe this in more detail here. We emphasize this discussion because i) it brings up a fundamental and so far unknown issue that might potentially exist in current approaches to explaining generalization and ii) rules it out before making more profound claims about uniform convergence.

Our argument specifically applies to margin-based Rademacher complexity approaches (such as Bartlett et al. [3], Neyshabur et al. [32]). These result in a bound like in Equation 1 that we recall here:

$$Pr_{(x,y)\sim\mathcal{D}}[\Gamma(f(\mathbf{x}),y) \leq 0] \leq \frac{1}{m} \sum_{(x,y)\in S} \mathbf{1}[\Gamma(f(\mathbf{x}),y) \leq \gamma]$$
$$+ \text{ generalization error bound.} \quad (1)$$

These methods upper bound the uniform convergence bound on the $\mathcal{L}^{(\gamma)}$ error on the network in terms of a uniform convergence bound on the margins of the network (see [25] for more details about margin theory of Rademacher complexity). The resulting generalization error bound in Equation 1 would take the following form, as per our notation from Definition 3.3:

$$\sup_{S\in\mathcal{S}_\delta} \sup_{h\in\mathcal{H}_\delta} \frac{1}{\gamma} \left| \mathbb{E}_\mathcal{D}[\Gamma(h(\mathbf{x}),y)] - \frac{1}{m} \sum_{(x,y)\in S} \Gamma(h(\mathbf{x}),y) \right|. \quad (2)$$

This particular upper bound on the generalization gap in the $\mathcal{L}^{(\gamma)}$ loss is also an upper bound on the generalization gap on the margins. That is, with high probability $1-\delta$ over the draws of $S$, the above bound is larger than the following term that corresponds to the difference in test/train margins:

$$\frac{1}{\gamma} \left( \mathbb{E}_{(x,y)\sim\mathcal{D}}[\Gamma(h_S(\mathbf{x}),y)] - \frac{1}{m} \sum_{(x,y)\in S} \Gamma(h_S(\mathbf{x}),y) \right). \quad (3)$$

We first argue that it is possible for the generalization error of the algorithm to decrease with $m$ (as roughly $m^{-0.5}$), but for the above quantity to be independent of $m$. As a result, the margin-based bound in Equation 2 (which is larger than Equation 3) will be non-decreasing in $m$, and even vacuous. Below we describe such a scenario.

Consider a network that first learns a simple hypothesis to fit the data, say, by learning a simple linear input-output mapping on linearly separable data. But subsequently, the classifier proceeds to *pseudo-overfit* to the samples by skewing up (down) the real-valued output of the network by some large constant $\Delta$ in a tiny neighborhood around the positive (negative) training inputs. Note that this would be possible if and only if the network is overparameterized. Now, even though the classifier's real-valued output is skewed around the training data, the decision boundary is still linear as the sign of the classifier's output has not changed on any input. Thus, the boundary is still simple and linear and the generalization error small.

However, the training margins are at least a constant $\Delta$ larger than the test margins (which are not affected by the bumps created in tiny regions around the training data). Then, the term in Equation 3 would be larger than $\Delta/\gamma$ and as a result, so would the term in Equation 2. Now in the generalization guarantee of Equation 1, recall that we must pick a value of $\gamma$ such that the first term is low i.e., most of the training datapoints must be classified by at least $\gamma$ margin. In this case, we can at best let $\gamma \approx \Delta$ as any larger value of $\gamma$ would make the margin-based training error non-negligible; as a result of this choice of $\gamma$, the bound in Equation 3 would be an $m$-independent constant close to 1. The same would also hold for its upper bound in Equation 2, which is the generalization bound provided by the margin-based techniques.

Clearly, this is a potential fundamental limitation in existing approaches, and if deep networks were indeed pseudo-overfitting this way, we would have identified the reason why at least some existing bounds are vacuous. However, (un)fortunately, we rule this out by observing that the difference in the train and test margins in Equation 3 does decrease with training dataset size $m$ (see Figure 12) as $\mathcal{O}(m^{-0.33})$. Additionally, this difference is numerically much less than $\gamma^\star = 10$ (which is the least margin by which 99% of the training data is classified) as long as $m$ is large, implying that Equation 3 is non-vacuous.

Figure 12: We plot the average margin of the network on the train and test data, and the difference between the two, the last of which decreases with $m$ as $\mathcal{O}(1/m^{0.33})$.

It is worth noting that the generalization error decreases at a faster rate of $\mathcal{O}(m^{-0.43})$ implying that the upper bound in Equation 3 which decreases only as $m^{-0.33}$, is loose. This already indicates a partial weakness in this specific approach to deriving generalization guarantees. Nevertheless, even this upper bound decreases at a significant rate with $m$ which the subsequent uniform convergence-based upper bound in Equation 2 is unable to capture, thus hinting at more fundamental weaknesses specific to uniform convergence.

**Do our example setups suffer from pseudo-overfitting?**   Before we wrap up this section, we discuss a question brought up by an anonymous reviewer, which we believe is worth addressing. Recall that in Section 3.1 and Section 3.2, we presented a linear and hypersphere classification task where we showed that uniform convergence provably fails. In light of the above discussion, one may be tempted to ask: do these two models fail to obey uniform convergence because of pseudo-overfitting?

The answer to this is that our proof for failure of uniform convergence in both these examples did not rely on any kind of pseudo-overfitting – had our proof relied on it, then we would have been able to show failure of only specific kinds of uniform convergence bounds (as discussed above). More formally, pseudo-overfitting in itself does not imply the lower bounds on $\epsilon_{\text{unif-alg}}$ that we have shown in these settings.

One may still be curious to understand the level of pseudo-overfitting in these examples, to get a sense of the similarity of this scenario with that of the MNIST setup. To this end, we note that our linear setup does indeed suffer from significant pseudo-overfitting – the classifier's output does indeed have bumps around each training point (which can be concluded from our proof).

In the case of the hypersphere example, we present Figure 13, where we plot of the average margins in this setup like in Figure 12. Here, we observe that, the mean margins on the test data (orange line) and on training data (blue line) do converge to each other with more training data size $m$ i.e., the gap in the mean test and training margins (green line) *does* decrease with $m$. Thus our setup exhibits a behavior similar to deep networks on MNIST in Figure 12. As noted in our earlier discussion, since the rate of decrease of the mean margin gap in MNIST is not as large as the decrease in test error itself, there should be "a small amount" of pseudo-overfitting in MNIST. The same holds in this setting, although, here we observe an even milder decrease, implying a larger amount of pseudo-overfitting. Nevertheless, we emphasize that, our proof shows that uniform convergence cannot capture even this decrease with $m$.

To conclude, pseudo-overfitting is certainly a phenomenon worth exploring better; however, our examples elucidate that there is a phenomenon beyond pseudo-overfitting that is at play in deep learning.

# D   Useful Lemmas

In this section, we state some standard results we will use in our proofs. We first define some constants: $c_1 = 1/2048$, $c_2 = \sqrt{15/16}$ and $c_3 = \sqrt{17/16}$ and $c_4 = \sqrt{2}$.

First, we state a tail bound for sub-exponential random variables [37].

Figure 13: In the hypersphere example of Section 3.2, we plot the average margin of the network on the train and test data, and the difference between the two. We observe the train and test margins do converge to each other.

**Lemma D.1.** *For a sub-exponential random variable $X$ with parameters $(\nu, b)$ and mean $\mu$, for all $t > 0$,*

$$Pr\left[|X - \mu| \geq t\right] \leq 2\exp\left(-\frac{1}{2}\min\left(\frac{t}{b}, \frac{t^2}{\nu^2}\right)\right).$$

As a corollary, we have the following bound on the sum of squared normal variables:

**Corollary D.1.1.** *For $z_1, z_2, \ldots, z_D \sim \mathcal{N}(0, 1)$, we have that*

$$Pr\left[\frac{1}{D}\sum_{j=1}^{D} z_j^2 \in [c_2^2, c_3^2]\right] \leq 2\exp(-c_1 D).$$

We now state the Hoeffding bound for sub-Gaussian random variable.

**Lemma D.2.** *Let $z_1, z_2, \ldots, z_D$ be independently drawn sub-Gaussian variables with mean $0$ and sub-gaussian parameter $\sigma_i$. Then,*

$$Pr\left[\left|\sum_{d=1}^{D} z_d\right| \geq t\right] \leq 2\exp(-t^2/2\sum_{d=1}^{D}\sigma_d^2).$$

Again, we restate it as follows:

**Corollary D.2.1.** *For any $\mathbf{u} = (u_1, u_2, \ldots, u_d) \in \mathbb{R}^D$, for $z_1, z_2, \ldots, z_D \sim \mathcal{N}(0, 1)$,*

$$Pr\left[\left|\sum_{d=1}^{D} u_d z_d\right| \geq \|\mathbf{u}\|_2 \cdot c_4\sqrt{\ln\frac{2}{\delta}}\right] \leq \delta.$$

# E   Proof for Theorem 3.1

In this section, we prove the failure of uniform convergence for our linear model. We first recall the setup:

**Distribution $\mathcal{D}$:** Each input $(\mathbf{x}_1, \mathbf{x}_2)$ is a $K + D$ dimensional vector where $\mathbf{x}_1 \in \mathbb{R}^K$ and $\mathbf{x}_2 \in \mathbb{R}^D$. $\mathbf{u} \in \mathbb{R}^K$ determines the centers of the classes. The label $y$ is drawn uniformly from $\{-1, +1\}$, and conditioned on $y$, we have $\mathbf{x}_1 = 2 \cdot y \cdot \mathbf{u}$ while $\mathbf{x}_2$ is sampled independently from $\mathcal{N}(0, \frac{32}{D}I)$.

**Learning algorithm $\mathcal{A}$:** We consider a linear classifier with weights $\mathbf{w} = (\mathbf{w}_1, \mathbf{w}_2)$. The output is computed as $h(\mathbf{x}) = \mathbf{w}_1\mathbf{x}_1 + \mathbf{w}_2\mathbf{x}_2$. Assume the weights are initialized to origin. Given $S = \{(\mathbf{x}^{(1)}, y^{(1)}), \ldots, (\mathbf{x}^{(m)}, y^{(m)})\}$, $\mathcal{A}$ takes a gradient step of learning rate 1 to maximize $y \cdot h(\mathbf{x})$ for each $(\mathbf{x}, y) \in S$. Regardless of the batch size, the learned weights would satisfy, $\mathbf{w}_1 = 2m\mathbf{u}$ and $\mathbf{w}_2 = \sum_i y^{(i)}\mathbf{x}_2^{(i)}$.

Below, we state the precise theorem statement (where we've used the constants $c_1 = 1/32$, $c_2 = 1/2$ and $c_3 = 3/2$ and $c_4 = \sqrt{2}$):

**Theorem 3.1** *In the setup above, for any $\epsilon, \delta > 0$ and $\delta < 1/4$, let $D$ be sufficiently large that it satisfies*

$$D \geq \frac{1}{c_1} \ln \frac{6m}{\delta}, \tag{4}$$

$$D \geq m \left( \frac{4c_4 c_3}{c_2^2} \right)^2 \ln \frac{6m}{\delta}, \tag{5}$$

$$D \geq m \left( \frac{4c_4 c_3}{c_2^2} \right)^2 \cdot 2\ln \frac{2}{\epsilon}, \tag{6}$$

*then we have that for all $\gamma \geq 0$, for the $\mathcal{L}^{(\gamma)}$ loss, $\epsilon_{\text{unif-alg}}(m, \delta) \geq 1 - \epsilon_{\text{gen}}(m, \delta)$.*

*Specifically, for $\gamma \in [0, 1]$, $\epsilon_{\text{gen}}(m, \delta) \leq \epsilon$, and so $\epsilon_{\text{unif-alg}}(m, \delta) \geq 1 - \epsilon$.*

*Proof.* The above follows from Lemma E.1, where we upper bound the generalization error, and from Lemma E.2 where we lower bound uniform convergence. $\qquad\square$

We first prove that the above algorithm generalizes well with respect to the losses corresponding to $\gamma \in [0, 1]$. First for the training data, we argue that both $\mathbf{w}_1$ and a small part of the noise vector $\mathbf{w}_2$ align along the correct direction, while the remaining part of the high-dimensional noise vector are orthogonal to the input; this leads to correct classification of the training set. Then, on the test data, we argue that $\mathbf{w}_1$ aligns well, while $\mathbf{w}_2$ contributes very little to the output of the classifier because it is high-dimensional noise. As a result, for most test data, the classification is correct, and hence the test and generalization error are both small.

**Lemma E.1.** *In the setup of Section 3, when $\gamma \in [0, 1]$, for $\mathcal{L}^{(\gamma)}$, $\epsilon_{\text{gen}}(m, \delta) \leq \epsilon$.*

*Proof.* The parameters learned by our algorithm satisfies $\mathbf{w}_1 = 2m \cdot \mathbf{u}$ and $\mathbf{w}_2 = \sum y^{(i)} \mathbf{x}_2^{(i)} \sim \mathcal{N}(0, \frac{8m}{c_2^2 D})$.

First, we have from Corollary D.1.1 that with probability $1 - \frac{\delta}{3m}$ over the draws of $\mathbf{x}_2^{(i)}$, as long as $\frac{\delta}{3m} \geq 2e^{-c_1 D}$ (which is given to hold by Equation 4),

$$c_2 \leq \frac{1}{2\sqrt{2}} c_2 \|\mathbf{x}_2^{(i)}\| \leq c_3. \tag{7}$$

Next, for a given $\mathbf{x}^{(i)}$, we have from Corollary D.2.1, with probability $1 - \frac{\delta}{3m}$ over the draws of $\sum_{j \neq i} y^{(j)} \mathbf{x}_2^{(j)}$,

$$|\mathbf{x}_2^{(i)} \cdot \sum_{j \neq i} y^{(j)} \mathbf{x}_2^{(j)}| \leq c_4 \|\mathbf{x}_2^{(i)}\| \frac{2\sqrt{2} \cdot \sqrt{m}}{c_2 \sqrt{D}} \sqrt{\ln \frac{6m}{\delta}}. \tag{8}$$

Then, with probability $1 - \frac{2}{3}\delta$ over the draws of the training dataset we have for all $i$,

$$y^{(i)} h(\mathbf{x}^{(i)}) = y^{(i)} \mathbf{w}_1 \cdot \mathbf{x}_1^{(i)} + y^{(i)} \cdot y^{(i)} \|\mathbf{x}_2^{(i)}\|^2 + y^{(i)} \cdot \mathbf{x}_2^{(i)} \cdot \sum_{j \neq i} y^{(j)} \mathbf{x}_2^{(j)}$$

$$= 4 + \underbrace{\|\mathbf{x}_2^{(i)}\|^2}_{\text{apply Equation 7}} + \underbrace{y^{(i)} \mathbf{x}_2^{(i)} \cdot \sum_{j \neq i} y^{(j)} \mathbf{x}_2^{(j)}}_{\text{apply Equation 8}}$$

$$\geq 4 + 4 \cdot 2 - c_4 \frac{2\sqrt{2} c_3}{c_2} \cdot \underbrace{\frac{2\sqrt{2} \cdot \sqrt{m}}{c_2 \sqrt{D}} \sqrt{\ln \frac{6m}{\delta}}}_{\text{apply Equation 5}}$$

$$\geq 4 + 8 - 2 = 10 > 1. \tag{9}$$

Thus, for all $\gamma \in [0, 1]$, the $\mathcal{L}^{(\gamma)}$ loss of this classifier on the training dataset $S$ is zero.

Now, from Corollary D.1.1, with probability $1 - \frac{\delta}{3}$ over the draws of the training data, we also have that, as long as $\frac{\delta}{3m} \geq 2e^{-c_1 D}$ (which is given to hold by Equation 4),

$$c_2 \sqrt{m} \leq \frac{1}{2\sqrt{2}} c_2 \| \sum y^{(i)} \mathbf{x}_2^{(i)} \| \leq c_3 \sqrt{m}. \tag{10}$$

Next, conditioned on the draw of $S$ and the learned classifier, for any $\epsilon' > 0$, with probability $1 - \epsilon'$ over the draws of a test data point, $(\mathbf{z}, y)$, we have from Corollary D.2.1 that

$$|\mathbf{z}_2 \cdot \sum y^{(i)} \mathbf{x}_2^{(i)}| \leq c_4 \| \sum y^{(i)} \mathbf{x}_2^{(i)} \| \cdot \frac{2\sqrt{2}}{c_2 \sqrt{D}} \cdot \ln \frac{1}{\epsilon'}. \tag{11}$$

Using this, we have that with probability $1 - 2\exp\left(-\frac{1}{2}\left(\frac{c_2^2}{4c_4 c_3} \sqrt{\frac{D}{m}}\right)^2\right)$ over the draws of a test data point, $(\mathbf{z}, y)$,

$$yh(\mathbf{x}) = y\mathbf{w}_1 \cdot \mathbf{z}_1 + \underbrace{y \cdot \mathbf{z}_2 \cdot \sum_j y^{(j)} \mathbf{x}_2^{(j)}}_{\text{apply Equation 11}}$$

$$\geq 4 - c_4 \underbrace{\| \sum y^{(i)} \mathbf{x}_2^{(i)} \|}_{\text{apply Equation 10}} \cdot \frac{2\sqrt{2}}{c_2 \sqrt{D}} \frac{c_2^2}{4c_4 c_3} \sqrt{\frac{D}{m}}$$

$$\geq 4 - 2 \geq 2. \tag{12}$$

Thus, we have that for $\gamma \in [0, 1]$, the $\mathcal{L}^{(\gamma)}$ loss of the classifier on the distribution $\mathcal{D}$ is $2\exp\left(-\frac{1}{2}\left(\frac{c_2^2}{4c_4 c_3} \sqrt{\frac{D}{m}}\right)^2\right)$ which is at most $\epsilon$ as assumed in Equation 6. In other words, the absolute difference between the distribution loss and the train loss is at most $\epsilon$ and this holds for at least $1 - \delta$ draws of the samples $S$. Then, by the definition of $\epsilon_{\text{gen}}$ we have the result.

$\square$

We next prove our uniform convergence lower bound. The main idea is that when the noise vectors in the training samples are negated, with high probability, the classifier misclassifies the training data. We can then show that for any choice of $\mathcal{S}_\delta$ as required by the definition of $\epsilon_{\text{unif-alg}}$, we can always find an $S_\star$ and its noise-negated version $S'_\star$ both of which belong to $\mathcal{S}_\delta$. Furthermore, we can show that $h_{S_\star}$ has small test error but high empirical error on $S'_\star$, and that this leads to a nearly vacuous uniform convergence bound.

**Lemma E.2.** *In the setup of Section 3, for any $\epsilon > 0$ and for any $\delta \leq 1/4$, and for the same lower bounds on $D$, and for any $\gamma \geq 0$, we have that*

$$\epsilon_{\text{unif-alg}}(m, \delta) \geq 1 - \epsilon_{\text{gen}}(m, \delta)$$

*for the $\mathcal{L}^{(\gamma)}$ loss.*

*Proof.* For any $S$, let $S'$ denote the set of noise-negated samples $S' = \{((\mathbf{x}_1, -\mathbf{x}_2), u) \mid ((\mathbf{x}_1, \mathbf{x}_2), y) \in S\}$. We first show with high probability $1 - 2\delta/3$ over the draws of $S$, that the classifier learned on $S$, misclassifies $S'$ completely. The proof for this is nearly identical to our proof for why the training loss is zero, except for certain sign changes. For any $\mathbf{x}_{\text{neg}}^{(i)} = (\mathbf{x}_1^{(i)}, -\mathbf{x}_2^{(i)})$, we have

$$y^{(i)} h(\mathbf{x}_{\text{neg}}^{(i)}) = y^{(i)} \mathbf{w}_1 \cdot \mathbf{x}_1^{(i)} - y^{(i)} \cdot y^{(i)} \| \mathbf{x}_2^{(i)} \|^2 - y^{(i)} \cdot \mathbf{x}_2^{(i)} \cdot \sum_{j \neq i} y^{(j)} \mathbf{x}_2^{(j)}$$

$$= 4 - \underbrace{\|\mathbf{x}_2^{(i)}\|^2}_{\text{apply Equation 7}} - \underbrace{y^{(i)}\mathbf{x}_2^{(i)} \cdot \sum_{j \neq i} y^{(j)}\mathbf{x}_2^{(j)}}_{\text{apply Equation 8}}$$

$$\leq 4 - 4 \cdot 2 + c_4 \frac{2\sqrt{2}c_3}{c_2} \cdot \underbrace{\frac{2\sqrt{2} \cdot \sqrt{m}}{c_2\sqrt{D}} \ln \frac{3m}{\delta}}_{\text{apply Equation 5}}$$

$$\leq 4 - 8 + 2 = -2 < 0.$$

Since the learned hypothesis misclassifies all of $S'$, it has loss of $1$ on $S'$.

Now recall that, by definition, to compute $\epsilon_{\text{unif-alg}}$, one has to pick a sample set space $\mathcal{S}_\delta$ of mass $1 - \delta$ i.e., $Pr_{S \sim \mathcal{S}^m}[S \in \mathcal{S}_\delta] \geq 1 - \delta$. We first argue that for *any* choice of $\mathcal{S}_\delta$, there must exist a 'bad' $S_\star$ such that (i) $S_\star \in \mathcal{S}_\delta$, (ii) $S'_\star \in \mathcal{S}_\delta$, (iii) $h_{S_\star}$ has test error less than $\epsilon_{\text{gen}}(m, \delta)$ and (iv) $h_{S_\star}$ completely misclassifies $S'_\star$.

We show the existence of such an $S_\star$, by arguing that over the draws of $S$, there is non-zero probability of picking an $S$ that satisfies all the above conditions. Specifically, we have by the union bound that

$$Pr_{S \sim \mathcal{D}^m} \left[ S \in \mathcal{S}_\delta, S' \in \mathcal{S}_\delta, \mathcal{L}_\mathcal{D}(h_S) \leq \epsilon_{\text{gen}}(m, \delta), \hat{\mathcal{L}}_{S'}(h_S) = 1 \right]$$
$$\geq 1 - Pr_{S \sim \mathcal{D}^m} [S \notin \mathcal{S}_\delta] - Pr_{S \sim \mathcal{D}^m} [S' \notin \mathcal{S}_\delta]$$
$$- Pr_{S \sim \mathcal{D}^m} [\mathcal{L}_\mathcal{D}(h_S) > \epsilon_{\text{gen}}(m, \delta)] - Pr_{S \sim \mathcal{D}^m} \left[ \hat{\mathcal{L}}_{S'}(h_S) \neq 1 \right]. \tag{13}$$

By definition of $\mathcal{S}_\delta$, we know $Pr_{S \sim \mathcal{D}^m} [S \notin \mathcal{S}_\delta] \leq \delta$. Similarly, by definition of the generalization error, we know that $Pr_{S \sim \mathcal{D}^m} [\mathcal{L}_\mathcal{D}(h_S) > \epsilon_{\text{gen}}(m, \delta)] \leq \delta$. We have also established above that $Pr_{S \sim \mathcal{D}^m} \left[ \hat{\mathcal{L}}_{S'}(h_S) \neq 1 \right] \leq 2\delta/3$. As for the term $Pr_{S \sim \mathcal{D}^m} [S' \notin \mathcal{S}_\delta]$, observe that under the draws of $S$, the distribution of the noise-negated dataset $S'$ is identical to $\mathcal{D}^m$. This is because the isotropic Gaussian noise vectors have the same distribution under negation. Hence, again by definition of $\mathcal{S}_\delta$, even this probability is at most $\delta$. Thus, we have that the probability in the left hand side of Equation 13 is at least $1 - 4\delta$, which is positive as long as $\delta < 1/4$.

This implies that for any given choice of $\mathcal{S}_\delta$, there exists $S_\star$ that satisfies our requirement. Then, from the definition of $\epsilon_{\text{unif-alg}}(m, \delta)$, we essentially have that,

$$\epsilon_{\text{unif-alg}}(m, \delta) = \sup_{S \in \mathcal{S}_\delta} \sup_{h \in \mathcal{H}_\delta} |\mathcal{L}_\mathcal{D}(h) - \hat{\mathcal{L}}_S(h)|$$
$$\geq |\mathcal{L}_\mathcal{D}(h_{S_\star}) - \hat{\mathcal{L}}_{S'_\star}(h)| = |\epsilon - 1| = 1 - \epsilon.$$

$\square$

## F   Neural Network with Exponential Activations

In this section, we prove the failure of uniform convergence for a neural network model with exponential activations. We first define the setup.

**Distribution**   Let $\mathbf{u}$ be an arbitrary vector in $D$ dimensional space such that $\|\mathbf{u}\| = \sqrt{D}/2$. Consider an input distribution in $2D$ dimensional space such that, conditioned on the label $y$ drawn from uniform distribution over $\{-1, +1\}$, the first $D$ dimensions $\mathbf{x}_1$ of a random point is given by $y\mathbf{u}$ and the remaining $D$ dimensions are drawn from $\mathcal{N}(0, 1)$. Note that in this section, we require $D$ to be only as large as $\ln m$, and not as large as $m$.

**Architecture.**   We consider an infinite width neural network with exponential activations, in which only the output layer weights are trainable. The hidden layer weights are frozen as initialized. Note that this is effectively a linear model with infinitely many randomized features. Indeed, recent work [17] has shown that under some conditions on how deep networks are initialized and parameterized, they behave a linear models on randomized features. Specifically, each hidden unit corresponds to a

distinct (frozen) weight vector $\mathbf{w} \in \mathbb{R}^{2D}$ and an output weight $a_{\mathbf{w}}$ that is trainable. We assume that the hidden layer weights are drawn from $\mathcal{N}(0, I)$ and $a_{\mathbf{w}}$ initialized to zero. Note that the output of the network is determined as

$$h(\mathbf{x}) = \mathbb{E}_{\mathbf{w}}[a_{\mathbf{w}} \exp(\mathbf{w} \cdot \mathbf{x})].$$

**Algorithm** We consider an algorithm that takes a gradient descent step to maximize $y \cdot h(\mathbf{x})$ for each $(\mathbf{x}, y)$ in the training dataset, with learning rate $\eta$. However, since, the function above is not a discrete sum of its hidden unit outputs, to define the gradient update on $a_{\mathbf{w}}$, we must think of $h$ as a functional whose input function maps every $\mathbf{w} \in \mathbb{R}^{2D}$ to $a_{\mathbf{w}} \in \mathbb{R}$. Then, by considering the functional derivative, one can conclude that the update on $a_{\mathbf{w}}$ can be written as

$$a_{\mathbf{w}} \leftarrow a_{\mathbf{w}} + \eta y \cdot \exp(\mathbf{w} \cdot \mathbf{x}) \cdot p(\mathbf{w}). \tag{14}$$

where $p(\mathbf{w})$ equals the p.d.f of $\mathbf{w}$ under the distribution it is drawn from. In this case $p(\mathbf{w}) = \frac{1}{(2\pi)^D} \exp\left(-\frac{\|\mathbf{w}\|^2}{2}\right)$.

In order to simplify our calculations we will set $\eta = (4\pi)^D$, although our analysis would extend to other values of the learning rate too. Similarly, our results would only differ by constants if we consider the alternative update rule, $a_{\mathbf{w}} \leftarrow a_{\mathbf{w}} + \eta y \cdot \exp(\mathbf{w} \cdot \mathbf{x})$.

We now state our main theorem (in terms of constants $c_1, c_2, c_3, c_4$ defined in Section D).

**Theorem F.1.** *In the set up above, for any $\epsilon, \delta > 0$ and $\delta < 1/4$, let $D$ and $m$ be sufficiently large that it satisfies*

$$D \geq \max\left(\frac{1}{c_2}, (16c_3c_4)^2\right) \cdot 2 \ln \frac{6m}{\epsilon} \tag{15}$$

$$D \geq \max\left(\frac{1}{c_2}, (16c_3c_4)^2\right) \cdot 2 \ln \frac{6m}{\delta} \tag{16}$$

$$D \geq 6 \ln 2m \tag{17}$$

$$m > \max 8 \ln \frac{6}{\delta}. \tag{18}$$

*then we have that for all $\gamma \geq 0$, for the $\mathcal{L}^{(\gamma)}$ loss, $\epsilon_{unif\text{-}alg}(m, \delta) \geq 1 - \epsilon_{gen}(m, \delta)$.*

*Specifically, for $\gamma \in [0, 1]$, $\epsilon_{gen}(m, \delta) \leq \epsilon$, and so $\epsilon_{unif\text{-}alg}(m, \delta) \geq 1 - \epsilon$.*

*Proof.* The result follows from the following lemmas. First in Lemma F.2, we derive the closed form expression for the function computed by the learned network. In Lemma F.3, we upper bound the generalization error and in Lemma F.4, we lower bound uniform convergence. $\square$

We first derive a closed form expression for how the output of the network changes under a gradient descent step on a particular datapoint.

**Lemma F.2.** *Let $h^{(0)}(\cdot)$ denote the function computed by the network before updating the weights. After updating the weights on a particular input $(\mathbf{x}, y)$ according to Equation 14, the learned network corresponds to:*

$$h(\mathbf{z}) = h^{(0)}(\mathbf{z}) + y \exp\left(\left\|\frac{\mathbf{z} + \mathbf{x}}{2}\right\|^2\right).$$

*Proof.* From equation 14, we have that

$$\frac{h(\mathbf{z}) - h^{(0)}(\mathbf{z})}{\eta}$$

$$= \int_{\mathbf{w}} (y \cdot \exp(\mathbf{w} \cdot \mathbf{x})p(\mathbf{w})) \cdot \exp(\mathbf{w} \cdot \mathbf{z})p(\mathbf{w})d\mathbf{w}$$

$$= y \int_{\mathbf{w}} \exp(\mathbf{w} \cdot (\mathbf{x} + \mathbf{z})) \cdot \left(\frac{1}{2\pi}\right)^{2D} \exp(-\|\mathbf{w}\|^2)d\mathbf{w}$$

$$= y \left(\frac{1}{2\pi}\right)^{2D} \int_{\mathbf{w}} \exp(\mathbf{w} \cdot (\mathbf{x} + \mathbf{z}) - \|\mathbf{w}\|^2)d\mathbf{w}$$

$$= y \left(\frac{1}{2\pi}\right)^{2D} \exp\left(\left\|\frac{\mathbf{z} + \mathbf{x}}{2}\right\|^2\right) \times \int_{\mathbf{w}} \exp\left(-\left\|\mathbf{w} - \frac{\mathbf{z} + \mathbf{x}}{2}\right\|^2\right) d\mathbf{w}$$

$$= y \left(\frac{1}{4\pi}\right)^{D} \exp\left(\left\|\frac{\mathbf{z} + \mathbf{x}}{2}\right\|^2\right) \times \left(\frac{1}{\sqrt{2\pi(0.5)}}\right)^{2D} \int_{\mathbf{w}} \exp\left(-\left\|\mathbf{w} - \frac{\mathbf{z} + \mathbf{x}}{2}\right\|^2\right) d\mathbf{w}$$

$$= y \left(\frac{1}{4\pi}\right)^{D} \exp\left(\left\|\frac{\mathbf{z} + \mathbf{x}}{2}\right\|^2\right).$$

In the last equality above, we make use of the fact that the second term corresponds to the integral of the p.d.f of $\mathcal{N}(\frac{\mathbf{z}+\mathbf{x}}{2}, 0.5I)$ over $\mathbb{R}^{2D}$. Since we set $\eta = (4\pi)^D$ gives us the final answer. □

Next, we argue that the generalization error of the algorithm is small. From Lemma F.2, we have that the output of the network is essentially determined by a summation of contributions from every training point. To show that the training error is zero, we argue that on any training point, the contribution from that training point dominates all other contributions, thus leading to correct classification. On any test point, we similarly show that the contribution of training points of the same class as that test point dominates the output of the network. Note that our result requires $D$ to scale only logarithmically with training samples $m$.

**Lemma F.3.** *In the setup of Section 3, when $\gamma \in [0, 1]$, for $\mathcal{L}^{(\gamma)}$, $\epsilon_{gen}(m, \delta) \leq \epsilon$.*

*Proof.* We first establish a few facts that hold with high probability over the draws of the training set $S$. First, from Corollary D.1.1 we have that, since $D \geq \frac{1}{c_2} \ln \frac{3m}{\delta}$ (from Equation 15), with probability at least $1 - \delta/3$ over the draws of $S$, for all $i$, the noisy part of each training input can be bounded as

$$c_2\sqrt{D} \leq \|\mathbf{x}_2^{(i)}\| \leq c_3\sqrt{D}. \tag{19}$$

Next, from Corollary D.2.1, we have that with probability at least $1 - \frac{\delta}{3m^2}$ over the draws of $\mathbf{x}_2^{(i)}$ and $\mathbf{x}_2^{(j)}$ for $i \neq j$,

$$|\mathbf{x}_2^{(i)} \cdot \mathbf{x}_2^{(j)}| \leq \|\mathbf{x}_2^{(i)}\| \cdot c_4\sqrt{2\ln\frac{6m}{\delta}}. \tag{20}$$

Then, by a union bound, the above two equations hold for all $i \neq j$ with probability at least $1 - \delta/2$.

Next, since each $y^{(i)}$ is essentially an independent sub-Gaussian with mean 0 and sub-Gaussian parameter $\sigma = 1$, we can apply Hoeffding's bound (Lemma D.2) to conclude that with probability at least $1 - \delta/3$ over the draws of $S$,

$$\left|\sum_{j=1}^{m} y^{(j)}\right| \leq \underbrace{\sqrt{2m\ln\frac{6}{\delta}}}_{Eq\ 18} < \frac{m}{2}. \tag{21}$$

Note that this means that there must exist at least one training data in each class.

Given these facts, we first show that the training error is zero by showing that for all $i$, $y^{(i)}h(\mathbf{x}^{(i)})$ is sufficiently large. On any training input $(\mathbf{x}^{(i)}, y^{(i)})$, using Lemma F.2, we can write

$$y^{(i)}h(\mathbf{x}^{(i)}) = \exp\left(\left\|\mathbf{x}^{(i)}\right\|^2\right) + \sum_{j\neq i} y^{(i)}y^{(j)}\exp\left(\left\|\frac{\mathbf{x}^{(i)}+\mathbf{x}^{(j)}}{2}\right\|^2\right)$$

$$\geq \exp\left(\left\|\mathbf{x}^{(i)}\right\|^2\right) - \sum_{\substack{j\neq i \\ y^{(i)}\neq y^{(j)}}} \exp\left(\left\|\frac{\mathbf{x}^{(i)}+\mathbf{x}^{(j)}}{2}\right\|^2\right)$$

$$\geq \exp\left(\left\|\mathbf{x}^{(i)}\right\|^2\right) \times \left(1 - \sum_{\substack{j\neq i \\ y^{(i)}\neq y^{(j)}}} \exp\left(\frac{\|\mathbf{x}^{(i)}+\mathbf{x}^{(j)}\|^2 - 4\|\mathbf{x}^{(i)}\|^2}{4}\right)\right).$$

Now, for any $j$ such that $y^{(j)} \neq y^{(i)}$, we have that

$$\|\mathbf{x}^{(i)}+\mathbf{x}^{(j)}\|^2 - 4\|\mathbf{x}^{(i)}\|^2 = -3\|\mathbf{x}^{(i)}\|^2 + \|\mathbf{x}_1^{(j)}\|^2 + 2\mathbf{x}_1^{(i)}\cdot\mathbf{x}_1^{(j)}$$

$$+ \underbrace{\|\mathbf{x}_2^{(j)}\|^2}_{Eq\ 19} + \underbrace{2\mathbf{x}_2^{(i)}\cdot\mathbf{x}_2^{(j)}}_{Eq\ 20}$$

$$\leq -3\|\mathbf{u}\|^2 - 3\underbrace{\|\mathbf{x}_2^{(i)}\|^2}_{Eq\ 19} + \|\mathbf{u}\|^2 - 2\|\mathbf{u}\|^2 + c_3^2 D + \underbrace{\|\mathbf{x}_2^{(i)}\|}_{Eq\ 19}\cdot 2c_4\sqrt{2\ln\frac{6m}{\delta}}$$

$$\leq -4\|\mathbf{u}\|^2 - 3c_2^2 D + c_3^2 D + \underbrace{\sqrt{D}\cdot c_3 c_4\sqrt{2\ln\frac{6m}{\delta}}}_{Eq\ 16}$$

$$\leq -1 - \frac{45}{16}D + \frac{17}{16}D + \frac{1}{16}D = -\frac{43}{16}D.$$

Plugging this back in the previous equation we have that

$$y^{(i)}h(\mathbf{x}^{(i)}) \geq \geq \exp\left(\underbrace{\left\|\mathbf{x}^{(i)}\right\|^2}_{Eq\ 19}\right)\left(1 - m\underbrace{\exp\left(-\frac{43}{64}D\right)}_{Eq\ 17}\right)$$

$$\geq \underbrace{\exp\left(\frac{15}{16}D\right)}_{Eq\ 17}\cdot\frac{1}{2} \geq 1.$$

Hence, $\mathbf{x}^{(i)}$ is correctly classified by a margin of 1 for every $i$.

Now consider any test data point $(\mathbf{z}, y)$. Since $D \geq \frac{1}{c_2}\ln\frac{2}{\epsilon}$ (Equation 16), we have that with probability at least $1 - \epsilon/2$ over the draws of $\mathbf{z}_2$, by Corollary D.1.1

$$c_2\sqrt{D} \leq \|\mathbf{z}_2\| \leq c_3\sqrt{D}. \tag{22}$$

Similarly, for each $i$, we have that with probability at least $1 - \epsilon/2m$ over the draws of $\mathbf{z}$, the following holds good by Corollary D.2.1

$$|\mathbf{x}_2^{(i)}\cdot\mathbf{z}_2| \leq \|\mathbf{x}_2^{(i)}\|\cdot c_4\sqrt{2\ln\frac{6m}{\epsilon}}. \tag{23}$$

Hence, the above holds over at least $1 - \epsilon/2$ draws of $\mathbf{z}$, and by extension, both the above equations hold over at least $1 - \epsilon$ draws of $\mathbf{z}$.

Now, for any $i$ such that $y^{(i)} = y$, we have that

$$\|\mathbf{x}^{(i)} + \mathbf{z}\|^2 = \|\mathbf{x}_1^{(i)}\|^2 + \|\mathbf{z}_1\|^2 + 2\mathbf{x}_1^{(i)} \cdot \mathbf{z}_1 + \underbrace{\|\mathbf{x}_2^{(i)}\|^2}_{Eq\ 19} + \underbrace{\|\mathbf{z}_2^{(i)}\|^2}_{Eq\ 22} + 2\underbrace{\mathbf{x}_2^{(i)} \cdot \mathbf{z}_2}_{Eq\ 23,\ 19}$$

$$\geq 4\|\mathbf{u}\|^2 + 2c_2^2 D - \underbrace{\sqrt{D} \cdot 2c_3 c_4 \sqrt{2 \ln \frac{6m}{\epsilon}}}_{Eq\ 15}$$

$$\geq D + \frac{30}{16}D - \frac{1}{16}D = \frac{45}{16}D.$$

Similarly, for any $i$ such that $y^{(i)} \neq y$, we have that

$$\|\mathbf{x}^{(i)} + \mathbf{z}\|^2 = \|\mathbf{x}_1^{(i)}\|^2 + \|\mathbf{z}_1\|^2 + 2\mathbf{x}_1^{(i)} \cdot \mathbf{z}_1 + \underbrace{\|\mathbf{x}_2^{(i)}\|^2}_{Eq\ 19} + \underbrace{\|\mathbf{z}_2^{(i)}\|^2}_{Eq\ 22} + 2\underbrace{\mathbf{x}_2^{(i)} \cdot \mathbf{z}_2}_{Eq\ 20,\ 19}$$

$$\leq 2\|\mathbf{u}\|^2 - 2\|\mathbf{u}\|^2 + 2c_3^2 D + \underbrace{\sqrt{D} \cdot 2c_3 c_4 \sqrt{2 \ln \frac{6m}{\delta}}}_{Eq\ 16}$$

$$\leq \frac{34}{16}D - \frac{1}{16}D = \frac{33}{16}D.$$

Since from Equation 21 we know there exists at least one training sample with a given label, we have that

$$y^{(i)} h(\mathbf{x}^{(i)}) = \sum_{i: y^{(i)} = y} \exp\left(\left\|\frac{\mathbf{x}^{(i)} + \mathbf{z}}{2}\right\|^2\right) - \sum_{i: y^{(i)} \neq y} \exp\left(\left\|\frac{\mathbf{x}^{(i)} + \mathbf{z}}{2}\right\|^2\right)$$

$$\geq \exp\left(\frac{45}{64}D\right) - m \exp\left(\frac{33}{64}D\right)$$

$$\geq \exp\left(\frac{45}{64}D\right) \cdot \left(1 - m \underbrace{\exp\left(-\frac{12}{64}D\right)}_{Eq\ 17}\right)$$

$$\geq \underbrace{\exp\left(\frac{45}{64}D\right)}_{Eq\ 17} \cdot \frac{1}{2} \geq 1.$$

Thus, at least $1 - \epsilon$ of the test datapoints are classified correctly.

$\square$

We next show that the uniform convergence bound is nearly vacuous. In order to do this, we create a set $S'$ from $S$ by negating all values but the noise vector. We then show that for every point in $S'$, the contribution from the corresponding point in $S$ dominates over the contribution from all other points. (This is because of how the non-negated noise vector in the point from $S'$ aligns adversarially with the noise vector from the corresponding point in $S$). As a result, the points in $S'$ are all labeled like in $S$, implying that $S'$ is completely misclassified. Then, similar to our previous arguments, we can show that uniform convergence is nearly vacuous.

**Lemma F.4.** *In the setup of Section F, for any $\epsilon > 0$ and for any $\delta \leq 1/4$, and for the same lower bounds on $D$ and $m$ as in Theorem F.1, and for any $\gamma \geq 0$, we have that*

$$\epsilon_{\text{unif-alg}}(m, \delta) \geq 1 - \epsilon_{\text{gen}}(m, \delta)$$

*for the $\mathcal{L}^{(\gamma)}$ loss.*

*Proof.* Let $S'$ be a modified version of the training set where all values are negated except that of the noise vectors i.e., $S' = \{((-\mathbf{x}_1, \mathbf{x}_2), -y) \mid ((\mathbf{x}_1, \mathbf{x}_2), y) \in S\}$. First we show that with probability at least $1 - 2\delta/3$ over the draws of $S$, $S'$ is completely misclassified. First, we have that with probability $1 - 2\delta/3$, Equations 19 and 20 hold good. Let $(\mathbf{x}_{\text{neg}}^{(i)}, y_{\text{neg}}^{(i)})$ denote the $i$th sample from $S'$. Then, we have that

$$
\begin{aligned}
y_{\text{neg}}^{(i)} h(\mathbf{x}_{\text{neg}}^{(i)}) &= -\exp\left(\left\|\frac{\mathbf{x}^{(i)} + \mathbf{x}_{\text{neg}}^{(i)}}{2}\right\|^2\right) + \sum_{j \neq i} y_{\text{neg}}^{(i)} y^{(j)} \exp\left(\left\|\frac{\mathbf{x}_{\text{neg}}^{(i)} + \mathbf{x}^{(j)}}{2}\right\|^2\right) \\
&\leq -\exp\left(\left\|\mathbf{x}_2^{(i)}\right\|^2\right) + \sum_{\substack{j \neq i \\ y_{\text{neg}}^{(i)} = y^{(j)}}} \exp\left(\left\|\frac{\mathbf{x}_{\text{neg}}^{(i)} + \mathbf{x}^{(j)}}{2}\right\|^2\right) \\
&\leq \exp\left(\left\|\mathbf{x}_2^{(i)}\right\|^2\right) \times \left(-1 + \sum_{\substack{j \neq i \\ y_{\text{neg}}^{(i)} = y^{(j)}}} \exp\left(\frac{\|\mathbf{x}_{\text{neg}}^{(i)} + \mathbf{x}^{(j)}\|^2 - 4\|\mathbf{x}_2^{(i)}\|^2}{4}\right)\right).
\end{aligned}
\tag{24}
$$

Now, consider $j$ such that $y^{(j)} = y_{\text{neg}}^{(i)}$. we have that

$$
\begin{aligned}
\|\mathbf{x}_{\text{neg}}^{(i)} + \mathbf{x}^{(j)}\|^2 - 4\|\mathbf{x}_2^{(i)}\|^2 &= \|\mathbf{x}_1^{(i)}\|^2 + \|\mathbf{x}_1^{(j)}\|^2 - 2\mathbf{x}_1^{(i)} \cdot \mathbf{x}_1^{(j)} - \underbrace{3\|\mathbf{x}_2^{(i)}\|^2}_{Eq\ 19} + \underbrace{\|\mathbf{x}_2^{(j)}\|^2}_{Eq\ 19} - \underbrace{2\mathbf{x}_2^{(i)} \cdot \mathbf{x}_2^{(j)}}_{Eq\ 20} \\
&\leq 4\|\mathbf{u}\|^2 - 3c_2^2 D + c_3 D + \underbrace{\|\mathbf{x}_2^{(i)}\|}_{Eq\ 19} \cdot 2c_4 \sqrt{2 \ln \frac{6m}{\delta}} \\
&\leq 4\|\mathbf{u}\|^2 - 3c_2^2 D + c_3^2 D + \underbrace{\sqrt{D} \cdot c_3 c_4 \sqrt{2 \ln \frac{6m}{\delta}}}_{Eq\ 16} \\
&\leq D - \frac{45}{16} D + \frac{17}{16} D + \frac{1}{16} D = \frac{-11}{16} D.
\end{aligned}
$$

Plugging the above back in Equation 24, we have

$$
\frac{y_{\text{neg}}^{(i)} h(\mathbf{x}_{\text{neg}}^{(i)})}{\exp\left(\left\|\mathbf{x}_2^{(i)}\right\|^2\right)} \leq -1 + m \underbrace{\exp\left(-11 D/64\right)}_{Eq\ 17} \leq -1/2,
$$

implying that $\mathbf{x}_{\text{neg}}^{(i)}$ is misclassified. This holds simultaneously for all $i$, implying that $S'$ is misclassified with high probability $1 - 2\delta/3$ over the draws of $S$. Furthermore, $S'$ has the same distribution as $\mathcal{D}^m$. Then, by the same argument as that of Lemma E.2, we can prove our final claim.

$\square$

# G   Further Remarks.

In this section, we make some clarifying remarks about our theoretical results.

### G.1 Nearly vacuous bounds for any $\gamma > 0$.

Typically, like in Mohri et al. [25], Bartlett et al. [3], the 0-1 test error is upper bounded in terms of the $\mathcal{L}^{(\gamma)}$ test error for some optimal choice of $\gamma > 0$ (as it is easier to apply uniform convergence for $\gamma > 0$). From the result in the main paper, it is obvious that for $\gamma \leq 1$, this approach would yield vacuous bounds. We now establish that this is the case even for $\gamma > 1$.

To help state this more clearly, for the scope of this particular section, let $\epsilon_{\text{unif-alg}}^{(\gamma)}, \epsilon_{\text{gen}}^{(\gamma)}$ denote the uniform convergence and generalization error for $\mathcal{L}^{(\gamma)}$ loss. Then, the following inequality is used to derive a bound on the 0-1 error:

$$\mathcal{L}_{\mathcal{D}}^{(0)}(h_S) \leq \mathcal{L}_{\mathcal{D}}^{(\gamma)}(h_S) \leq \hat{\mathcal{L}}_S^{(\gamma)}(h_S) + \epsilon_{\text{unif-alg}}^{(\gamma)}(m, \delta) \tag{25}$$

where the second inequality above holds with probability at least $1 - \delta$ over the draws of $S$, while the first holds for all $S$ (which follows by definition of $\mathcal{L}^{(\gamma)}$ and $\mathcal{L}^{(0)}$).

To establish that uniform convergence is nearly vacuous in any setting of $\gamma$, we must show that the right hand side of the above bound is nearly vacuous for any choice of $\gamma \geq 0$ (despite the fact that $\mathcal{L}_{\mathcal{D}}^{(0)}(S) \leq \epsilon$). In our results, we explicitly showed this to be true for only small values of $\gamma$, by arguing that the second term in the R.H.S, namely $\epsilon_{\text{unif-alg}}^{(\gamma)}(m, \delta)$, is nearly vacuous.

Below, we show that the above bound is indeed nearly vacuous for any value of $\gamma$, when we have that $\epsilon_{\text{unif-alg}}^{(\gamma)}(m, \delta) \geq 1 - \epsilon_{\text{gen}}^{(\gamma)}(m, \delta)$. Note that we established the relation $\epsilon_{\text{unif-alg}}^{(\gamma)}(m, \delta) \geq 1 - \epsilon_{\text{gen}}^{(\gamma)}(m, \delta)$ to be true in all of our setups.

**Proposition G.1.** *Given that for all $\gamma \geq 0$, $\epsilon_{\text{unif-alg}}^{(\gamma)}(m, \delta) \geq 1 - \epsilon_{\text{gen}}^{(\gamma)}(m, \delta)$ then, we then have that for all $\gamma \geq 0$,*

$$Pr_{S \sim \mathcal{D}^m}\left[\hat{\mathcal{L}}_S^{(\gamma)}(h_S) + \epsilon_{\text{unif-alg}}^{(\gamma)}(m, \delta) \geq \frac{1}{2}\right] > \delta$$

*or in other words, the guarantee from the right hand side of Equation 25 is nearly vacuous.*

*Proof.* Assume on the contrary that for some choice of $\gamma$, we are able to show that with probability at least $1 - \delta$ over the draws of $S$, the right hand side of Equation 25 is less than $1/2$. This means that $\epsilon_{\text{unif-alg}}^{(\gamma)}(m, \delta) < 1/2$. Furthermore, this also means that with probability at least $1 - \delta$ over the draws of $S$, $\hat{\mathcal{L}}_S^{(\gamma)}(h_S) < 1/2$ and $\mathcal{L}_{\mathcal{D}}^{(\gamma)}(h_S) < 1/2$ (which follows from the second inequality in Equation 25).

As a result, we have that with probability at least $1 - \delta$, $\mathcal{L}_{\mathcal{D}}^{(\gamma)}(h_S) - \hat{\mathcal{L}}_S^{(\gamma)}(h_S) < 1/2$. In other words, $\epsilon_{\text{gen}}^{(\gamma)}(m, \delta) < 1/2$. Since we are given that $\epsilon_{\text{unif-alg}}^{(\gamma)}(m, \delta) \geq 1 - \epsilon_{\text{gen}}^{(\gamma)}(m, \delta)$, by our upper bound on the generalization error, we have $\epsilon_{\text{unif-alg}}^{(\gamma)}(m, \delta) \geq 1/2$, which is a contradiction to our earlier inference that $\epsilon_{\text{unif-alg}}^{(\gamma)}(m, \delta) < 1/2$. Hence, our assumption is wrong.

$\square$

### G.2 Applicability of the observation in Section 3.2 to other settings

Recall that in the main paper, we discussed a setup where two hyperspheres of radius 1 and 1.1 respectively are classified by a sufficiently overparameterized ReLU network. We saw that even when the number of training examples was as large as 65536, we could project all of these examples on to the other corresponding hypersphere, to create a completely misclassified set $S'$. How well does this observation extend to other hyperparameter settings?

First, we note that in order to achieve full misclassification of $S'$, the network would have to be sufficiently overparameterized i.e., either the width or the input dimension must be larger. When the training set size $m$ is too large, one would observe that $S'$ is not as significantly misclassified as observed. (Note that on the other hand, increasing the parameter count would not hurt the generalization error. In fact it would improve it.)

Second, we note that our observation is sensitive to the choice of the difference in the radii between the hyperspheres (and potentially to other hyperparameters too). For example, when the outer sphere has radius 2, SGD learns to classify these spheres perfectly, resulting in zero error on both test data and on $S'$. As a result, our lower bound on $\epsilon_{\text{unif-alg}}$ would not hold in this setting.

However, here we sketch a (very) informal argument as to why there is reason to believe that our lower bound can still hold on a weaker notion of uniform convergence, a notion that is always applied in practice (in the main paper we focus on a strong notion of uniform convergence as a negative result about it is more powerful). More concretely, in reality, uniform convergence is computed without much knowledge about the data distribution, save a few weakly informative assumptions such as those bounding its support. Such a uniform convergence bound is effectively computed uniformly in supremum over a class of distributions.

Going back to the hypersphere example, the intuition is that even when the radii of the spheres are far apart, and hence, the classification perfect, the decision boundary learned by the network could still be microscopically complex – however these complexities are not exaggerated enough to misclassify $S'$. Now, for this given decision boundary, one would be able to construct an $S''$ which corresponds to projecting $S$ on two concentric hyperspheres that fall within these skews. Such an $S''$ would have a distribution that comes from some $\mathcal{D}'$ which, although not equal to $\mathcal{D}$, still obeys our assumptions about the underlying distribution. The uniform convergence bound which also holds for $\mathcal{D}'$ would thus have to be vacuous.

### G.3 On the dependence of $\epsilon_{\textbf{gen}}$ on $m$ in our examples.

As seen in the proof of Lemma E.1, the generalization error $\epsilon$ depends on $m$ and $D$ as $\mathcal{O}(e^{-D/m})$ ignoring some constants in the exponent. Clearly, this error decreases with the parameter count $D$.

On the other hand, one may also observe that this generalization error grows with the number of samples $m$, which might at first make this model seem inconsistent with our real world observations. However, we emphasize that this is a minor artefact of the simplifications in our setup, rather than a conceptual issue. With a small modification to our setup, we can make the generalization error decrease with $m$, mirroring our empirical observations. Specifically, in the current setup, we learn the true boundary along the first $K$ dimensions exactly. We can however modify it to a more standard learning setup where the boundary is not exactly recoverable and needs to be estimated from the examples. This would lead to an additional generalization error that scales as $\mathcal{O}(\sqrt{\frac{K}{m}})$ that is non-vacuous as long as $K \ll m$. Thus, the overall generalization error would be $\mathcal{O}(e^{-D/m} + \sqrt{\frac{K}{m}})$.

What about the overall dependence on $m$? Now, assume we have an overparameterization level of $D \gg m \ln(m/K)$, so that $e^{-D/m} \ll \sqrt{K/m}$. Hence, in the sufficiently overparameterized regime, the generalization error $\mathcal{O}(e^{-D/m})$ that comes from the noise we have modeled, pales in comparison with the generalization error that would stem from estimating the low-complexity boundary. Overall, as a function of $m$, the resulting error would behave like $\mathcal{O}(\sqrt{\frac{K}{m}})$ and hence show a decrease with increasing $m$ (as long the increase in $m$ is within the overparameterized regime).

### G.4 Failure of hypothesis-dependent uniform convergence bounds.

Often, uniform convergence bounds are written as a bound on the generalization error of a specific hypothesis rather than the algorithm. These bounds have an explicit dependence on the weights learned. As an example, a bound may be of the form that, with high probability over draws of training set $\tilde{S}$, for any hypothesis $h$ with weights $\mathbf{w}$,

$$\mathcal{L}_{\mathcal{D}}(h) - \hat{\mathcal{L}}_{\tilde{S}}(h) \leq \frac{\|\mathbf{w}\|_2}{\sqrt{m}}.$$

Below we argue why even these kinds of hypothesis-dependent bounds fail in our setting. We can informally define the tightest hypothesis-dependent uniform convergence bound as follows, in a

manner similar to Definition 3.3 of the tightest uniform convergence bound. Recall that we first pick a set of datasets $\mathcal{S}_\delta$ such that $Pr_{\tilde{S} \sim \mathcal{D}^m}[\tilde{S} \notin \mathcal{S}_\delta] \leq \delta$. Then, for all $\tilde{S} \in S_\delta$, we denote the upper bound on the generalization gap of $h_{\tilde{S}}$ by $\epsilon_{\text{unif-alg}}(h_{\tilde{S}}, m, \delta)$, where:

$$\epsilon_{\text{unif-alg}}(h_{\tilde{S}}, m, \delta) := \sup_{\tilde{S} \in \mathcal{S}_\delta} |\mathcal{L}_D(h_{\tilde{S}}) - \hat{\mathcal{L}}_S(h_{\tilde{S}})|.$$

In other words, the tightest upper bound here corresponds to the difference between the test and empirical error of the specific hypothesis $h_{\tilde{S}}$ but computed across nearly all datasets $S$ in $\mathcal{S}_\delta$.

To show failure of the above bound, recall from all our other proofs of failure of uniform convergence, we have that for at least $1 - O(\delta)$ draws of the sample set $\tilde{S}$, four key conditions are satisfied: (i) $\tilde{S} \in \mathcal{S}_\delta$, (ii) the corresponding bad dataset $\tilde{S}' \in \mathcal{S}_\delta$, (iii) the error on the bad set $\hat{\mathcal{L}}_{\tilde{S}'}(h_{\tilde{S}}) = 1$ and (iv) the test error $\mathcal{L}_D(h_{\tilde{S}}) \leq \epsilon_{\text{gen}}(m, \delta)$. For all such $\tilde{S}$, in the definition of $\epsilon_{\text{unif-alg}}(h_{\tilde{S}}, m, \delta)$, let us set $S$ to be $\tilde{S}'$. Then, we would get $\epsilon_{\text{unif-alg}}(h_{\tilde{S}}, m, \delta) \geq |\mathcal{L}_D(h_{\tilde{S}}) - \hat{\mathcal{L}}_{\tilde{S}'}(h_{\tilde{S}})| \geq 1 - \epsilon_{\text{gen}}(m, \delta)$. In other words, with probability at least $1 - O(\delta)$ over the draw of the training set, even a hypothesis-specific generalization bound fails to explain generalization of the corresponding hypothesis.

# H   An abstract setup

We now present an abstract setup that, although unconventional in some ways, conveys the essence behind how uniform convergence fails to explain generalization. Let the underlying distribution over the inputs be a spherical Gaussian in $\mathbb{R}^D$ where $D$ can be however small or large as the reader desires. Note that our setup would apply to many other distributions, but a Gaussian would make our discussion easier. Let the labels of the inputs be determined by some $h^\star : \mathbb{R}^D \to \{-1, +1\}$. Consider a scenario where the learning algorithm outputs a very slightly modified version of $h^\star$. Specifically, let $S' = \{-\mathbf{x} \mid \mathbf{x} \in S\}$; then, the learner outputs

$$h_S(\mathbf{x}) = \begin{cases} -h^\star(\mathbf{x}) & \text{if } \mathbf{x} \in S' \\ h^\star(\mathbf{x}) & \text{otherwise.} \end{cases}$$

That is, the learner misclassifies inputs that correspond to the negations of the samples in the training data – this would be possible if and only if the classifier is overparameterized with $\Omega(mD)$ parameters to store $S'$. We will show that uniform convergence fails to explain generalization for this learner.

First we establish that this learner generalizes well. Note that a given $S$ has zero probability mass under $\mathcal{D}$, and so does $S'$. Then, the training and test error are zero – except for pathological draws of $S$ that intersect with $S'$, which are almost surely never drawn from $\mathcal{D}^m$ – and hence, the generalization error of $\mathcal{A}$ is zero too.

It might thus seem reasonable to expect that one could explain this generalization using implicit-regularization-based uniform convergence by showing $\epsilon_{\text{unif-alg}}(m, \delta) = 0$. Surprisingly, this is not the case as $\epsilon_{\text{unif-alg}}(m, \delta)$ is in fact 1!

First it is easy to see why the looser bound $\epsilon_{\text{unif}}(m, \delta)$ equals 1, if we let $\mathcal{H}$ be the space of all hypotheses the algorithm could output: there must exist a non-pathological $S \in \mathcal{S}_\delta$, and we know that $h_{S'} \in \mathcal{H}$ misclassifies the negation of its training set, namely $S$. Then, $\sup_{h \in \mathcal{H}} |\mathcal{L}_\mathcal{D}(h) - \hat{\mathcal{L}}_S(h)| = |\mathcal{L}_\mathcal{D}(h_{S'}) - \hat{\mathcal{L}}_S(h_{S'})| = |0 - 1| = 1$.

One might hope that in the stronger bound of $\epsilon_{\text{unif-alg}}(m, \delta)$ since we truncate the hypothesis space, it is possible that the above adversarial situation would fall apart. However, with a more nuanced argument, we can similarly show that $\epsilon_{\text{unif-alg}}(m, \delta) = 1$. First, recall that any bound on $\epsilon_{\text{unif-alg}}(m, \delta)$, would have to pick a truncated sample set space $\mathcal{S}_\delta$. Consider any choice of $\mathcal{S}_\delta$, and the corresponding set of explored hypotheses $\mathcal{H}_\delta$. We will show that for any choice of $\mathcal{S}_\delta$, there exists $S_\star \in \mathcal{S}_\delta$ such that (i) $h_{S_\star}$ has zero test error and (ii) the negated training set $S'_\star$ belongs to $\mathcal{S}_\delta$ and (iii) $h_{S_\star}$ has error 1 on $S_\star$. Then, it follows that $\epsilon_{\text{unif-alg}}(m, \delta) = \sup_{S \in \mathcal{S}_\delta} \sup_{h \in \mathcal{H}_\delta} |\mathcal{L}_\mathcal{D}(h) - \hat{\mathcal{L}}_S(h)| \geq |\mathcal{L}_\mathcal{D}(h_{S_\star}) - \hat{\mathcal{L}}_{S'_\star}(h)| = |0 - 1| = 1$.

We can prove the existence of such an $S_\star$ by showing that the probability of picking one such set under $\mathcal{D}^m$ is non-zero for $\delta < 1/2$. Specifically, under $S \sim \mathcal{D}^m$, we have by the union bound that

$$Pr\left[\mathcal{L}_\mathcal{D}(h_S) = 0, \hat{\mathcal{L}}_{S'}(h_S) = 1, S \in \mathcal{S}_\delta, S' \in \mathcal{S}_\delta\right] \geq$$

$$1 - Pr\left[\mathcal{L}_\mathcal{D}(h_S) \neq 0, \hat{\mathcal{L}}_{S'}(h_S) \neq 1\right] - Pr\left[S \notin \mathcal{S}_\delta\right] - Pr\left[S' \notin \mathcal{S}_\delta\right].$$

Since the pathological draws have probability zero, the first probability term on the right hand side is zero. The second term is at most $\delta$ by definition of $\mathcal{S}_\delta$. Crucially, the last term too is at most $\delta$ because $S'$ (which is the negated version of $S$) obeys the same distribution as $S$ (since the isotropic Gaussian is invariant to a negation). Thus, the above probability is at least $1 - 2\delta > 0$, implying that there exist (many) $S_\star$, proving our main claim.

**Remark.** While our particular learner might seem artificial, much of this artificiality is only required to make the argument simple. The crucial trait of the learner that we require is that the misclassified region in the input space (i) covers low probability and yet (ii) is complex and highly dependent on the training set draw. Our intuition is that SGD-trained deep networks possess these traits.

# I Learnability and Uniform Convergence

Below, we provide a detailed discussion on learnability, uniform convergence and generalization. Specifically, we argue why the fact that uniform convergence is necessary for learnability does not preclude the fact that uniform convergence maybe unable to *explain* generalization of a particular algorithm for a particular distribution.

We first recall the notion of learnability. First, formally, a binary classification problem consists of a hypothesis class $\mathcal{H}$ and an instance space $\mathcal{X} \times \{-1, 1\}$. The problem is said to be *learnable* if there exists a learning rule $\mathcal{A}' : \bigcup_{m=1}^{\infty} \mathcal{Z}^m \to \mathcal{H}$ and a monotonically decreasing sequence $\epsilon_{\text{lnblty}}(m)$ such that $\epsilon_{\text{lnblty}}(m) \xrightarrow{m \to \infty} 0$ and

$$\forall \mathcal{D}' \ \mathbb{E}_{S \sim \mathcal{D}'^m}\left[\mathcal{L}_{\mathcal{D}'}^{(0)}(\mathcal{A}'(S)) - \min_{h \in \mathcal{H}} \mathcal{L}_{\mathcal{D}'}^{(0)}(h)\right] \leq \epsilon_{\text{lnblty}}(m). \tag{26}$$

Vapnik and Chervonenkis [36] showed that finite VC dimension of the hypothesis class is necessary and sufficient for learnability in binary classification problems. As Shalev-Shwartz et al. [34] note, since finite VC dimension is equivalent to uniform convergence, it can thus be concluded that uniform convergence is necessary and sufficient for learnability binary classification problems.

However, learnability is a strong notion that does not necessarily have to hold for a particular learning algorithm to generalize well for a particular underlying distribution. Roughly speaking, this is because learnability evaluates the algorithm under all possible distributions, including many complex distributions; while a learning algorithm may generalize well for a particular distribution under a given hypothesis class, it may fail to do so on more complex distributions under the same hypothesis class.

For more intuition, we present a more concrete but informal argument below. However, this argument is technically redundant because learnability is equivalent to uniform convergence for binary classification, and since we established the lack of necessity of uniform convergence, we effectively established the same for learnability too. However, we still provide the following informal argument as it provides a different insight into why learnability and uniform convergence are not necessary to explain generalization.

Our goal is to establish that in the set up of Section 3, even if we considered the binary classification problem corresponding to $\mathcal{H}_\delta$ (the class consisting of only those hypotheses explored by the algorithm $\mathcal{A}$ under a distribution $\mathcal{D}$), the corresponding binary classification problem is not learnable i.e., Equation 26 does not hold when we plug in $\mathcal{H}_\delta$ in place of $\mathcal{H}$.

First consider distributions of the following form that is more complex than the linearly separable $\mathcal{D}$: for any dataset $S'$, let $\mathcal{D}_{S'}$ be the distribution that has half its mass on the part of the linearly separable

distribution $\mathcal{D}$ excluding $S'$, and half its mass on the distribution that is uniformly distributed over $S'$. Now let $S'$ be a random dataset drawn from $\mathcal{D}$ *but with all its labels flipped*; consider the corresponding complex distribution $\mathcal{D}_{S'}$.

We first show that there exists $h \in \mathcal{H}_\delta$ that fits this distribution well. Now, for most draws of the "wrongly" labeled $S'$, we can show that the hypothesis $h$ for which $\mathbf{w}_1 = 2 \cdot \mathbf{u}$ and $\mathbf{w}_2 = \sum_{(x,y)\in S'} y \cdot \mathbf{x}_2$ fits the "wrong" labels of $S'$ perfectly; this is because, just as argued in Lemma E.2, $\mathbf{w}_2$ dominates the output on all these inputs, although $\mathbf{w}_1$ would be aligned incorrectly with these inputs. Furthermore, since $\mathbf{w}_2$ does not align with most inputs from $\mathcal{D}$, by an argument similar to Lemma E.1, we can also show that this hypothesis has at most $\epsilon$ error on $\mathcal{D}$, and that this hypothesis belongs to $\mathcal{H}_\delta$. Overall this means that, w.h.p over the choice of $S'$, there exists a hypothesis $h \in \mathcal{H}_\delta$ for which the error on the complex distribution $\mathcal{D}_{S'}$ is at most $\epsilon/2$ i.e.,

$$\min_{h \in \mathcal{H}} \mathbb{E}_{(x,y)\sim\mathcal{D}_{S'}}[\mathcal{L}(h(x),y)] \leq \epsilon/2.$$

On the other hand, let $\mathcal{A}'$ be any learning rule which outputs a hypothesis given $S \sim \mathcal{D}_{S'}$. With high probability over the draws of $S \sim \mathcal{D}_{S'}$, only at most, say $3/4$th of $S$ (i.e., $0.75m$ examples) will be sampled from $S'$ (and the rest from $\mathcal{D}$). Since the learning rule which has access only to $S$, has not seen at least a quarter of $S'$, with high probability over the random draws of $S'$, the learning rule will fail to classify roughly half of the unseen examples from $S'$ correctly (which would be about $(m/4) \cdot 1/2 = m/8$). Then, the error on $\mathcal{D}_{S'}$ will be at least $1/16$. From the above arguments, we have that $\epsilon_{\text{learnability}}(m) \geq 1/16 - \epsilon/2$, which is a non-negligible constant that is independent of $m$.

## J  Deterministic PAC-Bayes bounds are two-sided uniform convergence bounds

By definition, VC-dimension, Rademacher complexity and other covering number based bounds are known to upper bound the term $\epsilon_{\text{unif-alg}}$ and therefore our negative result immediately applies to all these bounds. However, it may not be immediately clear if bounds derived through the PAC-Bayesian approach fall under this category too. In this discussion, we show that existing deterministic PAC-Bayes based bounds are in fact two-sided in that they are lower bounded by $\epsilon_{\text{unif-alg}}$ too.

For a given prior distribution $P$ over the parameters, a PAC-Bayesian bound is of the following form: with high probability $1 - \delta$ over the draws of the data $S$, we have that *for all distributions $Q$* over the hypotheses space:

$$KL\left(\mathbb{E}_{\tilde{h}\sim Q}[\hat{\mathcal{L}}_S(\tilde{h})] \,\middle\|\, \mathbb{E}_{\tilde{h}\sim Q}[\mathcal{L}_\mathcal{D}(\tilde{h})]\right) \leq \underbrace{\frac{KL(Q\|P) + \ln\frac{2m}{\delta}}{m-1}}_{:=\epsilon_{\text{pb}}(P,Q,m,\delta)}. \tag{27}$$

Note that here for any $a, b \in [0,1]$, $KL(a\|b) = a\ln\frac{a}{b} + (1-a)\ln\frac{1-a}{1-b}$. Since the precise form of the PAC-Bayesian bound on the right hand side is not relevant for the rest of the discussion, we will concisely refer to it as $\epsilon_{\text{pb}}(P, Q, m, \delta)$. What is of interest to us is the fact that the above bound holds for all $Q$ for most draws of $S$ and that the KL-divergence on the right-hand side is in itself two-sided, in some sense.

Typically, the above bound is simplified to derive the following one-sided bound on the difference between the expected and empirical errors of a stochastic network (see [24] for example):

$$\mathbb{E}_{\tilde{h}\sim Q}[\mathcal{L}_\mathcal{D}(\tilde{h})] - \mathbb{E}_{\tilde{h}\sim Q}[\hat{\mathcal{L}}_S(\tilde{h})] \leq \sqrt{2\epsilon_{\text{pb}}(P,Q,m,\delta)} + 2\epsilon_{\text{pb}}(P,Q,m,\delta). \tag{28}$$

This bound is then manipulated in different ways to obtain bounds on the deterministic network. In the rest of this discussion, we focus on the two major such derandomizing techniques and argue that both these techniques boil down to two-sided convergence. While, we do not formally establish that there may exist other techniques which ensure that the resulting deterministic bound is strictly one-sided, we suspect that no such techniques may exist. This is because the KL-divergence bound in Equation 27 is in itself two-sided in the sense that for the right hand side bound to be small, both the stochastic test and train errors must be close to each other; it is not sufficient if the stochastic test error is smaller than the stochastic train error.

## J.1 Deterministic PAC-Bayesian Bounds of Type A

To derive a deterministic generalization bound, one approach is to add extra terms that account for the perturbation in the loss of the network [30, 24, 27]. That is, define:

$$\Delta(h, Q, \mathcal{D}) = |\mathcal{L}_{\mathcal{D}}(h) - \mathbb{E}_{\tilde{h} \sim Q}[\mathcal{L}_{\mathcal{D}}(\tilde{h})]|,$$

$$\Delta(h, Q, S) = \left| \hat{\mathcal{L}}_S(h) - \mathbb{E}_{\tilde{h} \sim Q}[\hat{\mathcal{L}}_S(\tilde{h})] \right|.$$

Then, one can get a deterministic upper bound as:

$$\mathcal{L}_{\mathcal{D}}(h) - \hat{\mathcal{L}}_S(h) \leq \sqrt{2\epsilon_{\text{pb}}(P, Q_h, m, \delta)} + 2\epsilon_{\text{pb}}(P, Q_h, m, \delta) + \Delta(h, Q, \mathcal{D}) + \Delta(h, Q, S).$$

Note that while applying this technique, for any hypothesis $h$, one picks a posterior $Q_h$ specific to that hypothesis (typically, centered at that hypothesis).

We formally define the deterministic bound resulting from this technique below. We consider the algorithm-dependent version and furthermore, we consider a bound that results from the best possible choice of $Q_h$ for all $h$. We define this deterministic bound in the format of $\epsilon_{\text{unif-alg}}$ as follows:

**Definition J.1.** The distribution-dependent, algorithm-dependent, deterministic PAC-Bayesian bound of (the hypothesis class $\mathcal{H}$, algorithm $\mathcal{A}$)-pair with respect to $\mathcal{L}$ is defined to be the smallest value $\epsilon_{\text{pb-det-A}}(m, \delta)$ such that the following holds:

1. there exists a set of $m$-sized samples $\mathcal{S}_\delta \subseteq (\mathcal{X} \times \{-1, +1\})^m$ for which:

$$Pr_{S \sim \mathcal{D}^m}[S \notin \mathcal{S}_\delta] \leq \delta,$$

2. and if we define $\mathcal{H}_\delta = \bigcup_{S \in \mathcal{S}_\delta} \{h_S\}$ to be the space of hypotheses explored only on these samples, then there must exist a prior $P$ and for each $h \in \mathcal{H}_\delta$, a distribution $Q_h$, such that uniform convergence must hold as follows:

$$\sup_{S \in \mathcal{S}_\delta} \sup_{h \in \mathcal{H}_\delta} \sqrt{2\epsilon_{\text{pb}}(P, Q_h, m, \delta)} + 2\epsilon_{\text{pb}}(P, Q_h, m, \delta)$$
$$+ \Delta(h, Q_h, \mathcal{D}) + \Delta(h, Q_h, S) < \epsilon_{\text{pb-det-A}}(m, \delta), \tag{29}$$

   as a result of which, by Equation 28, the following one-sided uniform convergence also holds:

$$\sup_{S \in \mathcal{S}_\delta} \sup_{h \in \mathcal{H}_\delta} \mathcal{L}_{\mathcal{D}}(h) - \hat{\mathcal{L}}_S(h) < \epsilon_{\text{pb-det-A}}(m, \delta). \tag{30}$$

Now, recall that $\epsilon_{\text{unif-alg}}(m, \delta)$ is a two-sided bound, and in fact our main proof crucially depended on this fact in order to lower bound $\epsilon_{\text{unif-alg}}(m, \delta)$. Hence, to extend our lower bound to $\epsilon_{\text{pb-det-A}}(m, \delta)$ we need to show that it is also two-sided in that it is lower bounded by $\epsilon_{\text{unif-alg}}(m, \delta)$. The following result establishes this:

**Theorem J.1.** *Let $\mathcal{A}$ be an algorithm such that on at least $1 - \delta$ draws of the training dataset $S$, the algorithm outputs a hypothesis $h_S$ that has $\hat{\epsilon}(m, \delta)$ loss on the training data $S$. Then*

$$e^{-3/2} \cdot \epsilon_{\text{unif-alg}}(m, 3\delta) - (1 - e^{-3/2})(\hat{\epsilon}(m, \delta) + \epsilon_{\text{gen}}(m, \delta)) \leq \epsilon_{\text{pb-det-A}}(m, \delta).$$

*Proof.* First, by the definition of the generalization error, we know that with probability at least $1 - \delta$ over the draws of $S$,

$$\mathcal{L}_D(h_S) \leq \hat{\mathcal{L}}_S(h_S) + \epsilon_{\text{gen}}(m, \delta).$$

Furthermore since the training loss it at most $\hat{\epsilon}(m, \delta)$ on at least $1 - \delta$ draws we have that on at least $1 - 2\delta$ draws of the dataset,

$$\mathcal{L}_D(h_S) \leq \hat{\epsilon}(m, \delta) + \epsilon_{\text{gen}}(m, \delta).$$

Let $\mathcal{H}_\delta$ and $\mathcal{S}_\delta$ be the subset of hypotheses and sample sets as in the definition of $\epsilon_{\text{pb-det-A}}$. Then, from the above, there exist $\mathcal{H}_{3\delta} \subseteq \mathcal{H}_\delta$ and $\mathcal{S}_{3\delta} \subseteq \mathcal{S}_\delta$ such that

$$Pr_{S \sim \mathcal{D}^m}[S \notin \mathcal{S}_{3\delta}] \leq 3\delta$$

and $\mathcal{H}_{3\delta} = \bigcup_{S \in \mathcal{S}_{3\delta}} \{h_S\}$, and furthermore,

$$\sup_{h \in \mathcal{H}_{3\delta}} \mathcal{L}_{\mathcal{D}}(h) \leq \hat{\epsilon}(m, \delta) + \epsilon_{\text{gen}}(m, \delta).$$

.

Using the above, and the definition of $\Delta$, we have for all $h \in \mathcal{H}_{3\delta}$, the following upper bound on its stochastic test error:

$$\mathbb{E}_{\tilde{h} \sim Q_h}[\mathcal{L}_{\mathcal{D}}(\tilde{h})] \leq \mathcal{L}_{\mathcal{D}}(h) + \Delta(h, Q_h, \mathcal{D}) \leq \hat{\epsilon}(m, \delta) + \epsilon_{\text{gen}}(m, \delta) + \underbrace{\Delta(h, Q_h, \mathcal{D})}_{\text{applying Equation 29}}$$

$$\leq \hat{\epsilon}(m, \delta) + \epsilon_{\text{gen}}(m, \delta) + \epsilon_{\text{pb-det-A}}(m, \delta). \tag{31}$$

.

Now, for each pair of $h \in \mathcal{H}_{3\delta}$ and $S \in \mathcal{S}_{3\delta}$, we will bound its empirical error minus the expected error in terms of $\epsilon_{\text{pb-det-A}}(m, \delta)$. For convenience, let us denote by $a := \mathbb{E}_{\tilde{h} \sim Q_h}[\hat{\mathcal{L}}_S(\tilde{h})]$ and $b := \mathbb{E}_{\tilde{h} \sim Q_h}[\mathcal{L}_{\mathcal{D}}(\tilde{h})]$ (note that $a$ and $b$ are terms that depend on a hypothesis $h$ and a sample set $S$).

We consider two cases. First, for some $h \in \mathcal{H}_{3\delta}$ and $S \in \mathcal{S}_{3\delta}$, consider the case that $e^{3/2}b > a$. Then, we have

$$\hat{\mathcal{L}}_S(h) - \mathcal{L}_{\mathcal{D}}(h) \leq a - b + \underbrace{\Delta(h, Q_h, \mathcal{D}) + \Delta(h, Q_h, S)}_{\text{applying Equation 29}}$$

$$\leq (e^{3/2} - 1) \underbrace{b}_{\text{apply Equation 31}} + \epsilon_{\text{pb-det-A}}(m, \delta)$$

$$\leq (e^{3/2} - 1)(\hat{\epsilon}(m, \delta) + \epsilon_{\text{gen}}(m, \delta) + \epsilon_{\text{pb-det-A}}(m, \delta))$$
$$+ \epsilon_{\text{pb-det-A}}(m, \delta)$$

$$\leq (e^{3/2} - 1)(\hat{\epsilon}(m, \delta) + \epsilon_{\text{gen}}(m, \delta)) + e^{3/2} \cdot \epsilon_{\text{pb-det-A}}(m, \delta). \tag{32}$$

Now consider the case where $a > e^{3/2}b$. This means that $(1 - a) < (1 - b)$. Then, if we consider the PAC-Bayesian bound of Equation 27,

$$a \ln \frac{a}{b} + (1 - a) \ln \frac{1 - a}{1 - b} \leq \epsilon_{\text{pb}}(P, Q_h, m, \delta), \tag{33}$$

on the second term, we can apply the inequality $\ln x \geq \frac{(x-1)(x+1)}{2x} = \frac{1}{2}\left(x - \frac{1}{x}\right)$ which holds for $x \in [0, 1]$ to get:

$$(1 - a) \ln \frac{1 - a}{1 - b} \geq \frac{1}{2}(1 - a)\left(\frac{1 - a}{1 - b} - \frac{1 - b}{1 - a}\right) = \left(\frac{(b - a)(2 - a - b)}{2(1 - b)}\right)$$

$$\geq -(a - b)\left(\frac{(2 - a - b)}{2(1 - b)}\right) \geq -(a - b)\left(\frac{(2 - b)}{2(1 - b)}\right)$$

$$\geq -\frac{(a - b)}{2}\left(\frac{1}{(1 - b)} + 1\right).$$

Plugging this back in Equation 33, we have,

$$\epsilon_{\text{pb}}(P, Q_h, m, \delta) \geq a \underbrace{\ln \frac{a}{b}}_{\geq 3/2} - \frac{(a - b)}{2}\left(\frac{1}{(1 - b)} + 1\right)$$

$$\geq \frac{2a(1-b)-(a-b)}{2(1-b)} + \frac{b}{2}$$

$$\geq \frac{2a(1-b)-(a-b)}{2(1-b)} \geq \frac{a-2ab+b}{2(1-b)}$$

$$\geq \frac{a-2ab+ab}{2(1-b)} \geq \frac{a}{2} \geq \frac{a-b}{2}$$

$$\geq \frac{1}{2}\left(\hat{\mathcal{L}}_S(h) - \mathcal{L}_{\mathcal{D}}(h) - (\Delta(h, Q_h, \mathcal{D}) + \Delta(h, Q_h, S))\right).$$

Rearranging, we get:

$$\hat{\mathcal{L}}_S(h) - \mathcal{L}_{\mathcal{D}}(h) \leq \underbrace{2\epsilon_{\text{pb}}(P, Q_h, m, \delta) + (\Delta(h, Q_h, \mathcal{D}) + \Delta(h, Q_h, S))}_{\text{Applying Equation 29}}$$

$$\leq \epsilon_{\text{pb-det-A}}(m, \delta). \tag{34}$$

Since, for all $h \in \mathcal{H}_{3\delta}$ and $S \in \mathcal{S}_{3\delta}$, one of Equations 32 and 34 hold, we have that:

$$\frac{1}{e^{3/2}}\left(\sup_{h \in \mathcal{H}_{3\delta}} \sup_{S \in \mathcal{S}_{3\delta}} \hat{\mathcal{L}}_S(h) - \mathcal{L}_{\mathcal{D}}(h)\right) - \frac{(e^{3/2}-1)}{e^{3/2}}(\hat{\epsilon}(m, \delta) + \epsilon_{\text{gen}}(m, \delta)) \leq \epsilon_{\text{pb-det-A}}(m, \delta).$$

It follows from Equation 30 that the above bound holds good even after we take the absolute value of the first term in the left hand side. However, the absolute value is lower-bounded by $\epsilon_{\text{unif-alg}}(m, 3\delta)$ (which follows from how $\epsilon_{\text{unif-alg}}(m, 3\delta)$ is defined to be the smallest possible value over the choices of $\mathcal{H}_{3\delta}, \mathcal{S}_{3\delta}$).

$\square$

As a result of the above theorem, we can show that $\epsilon_{\text{pb-det-A}}(m, \delta) = \Omega(1) - \mathcal{O}(\epsilon)$, thus establishing that, for sufficiently large $D$, even though the generalization error would be negligibly small, the PAC-Bayes based bound would be as large as a constant.

**Corollary J.1.1.** *In the setup of Section 3, for any $\epsilon, \delta > 0, \delta < 1/12$, when $D = \Omega\left(\max\left(m \ln \frac{3}{\delta}, m \ln \frac{1}{\epsilon}\right)\right)$, we have,*

$$e^{-3/2} \cdot (1-\epsilon) - (1-e^{-3/2})(\epsilon) \leq \epsilon_{pb\text{-}det\text{-}A}(m, \delta).$$

*Proof.* The fact that $\epsilon_{\text{gen}}(m, \delta) \leq \epsilon$ follows from Theorem 3.1. Additionally, $\hat{\epsilon}(m, \delta) = 0$ follows from the proof of Theorem 3.1. Now, as long as $3\delta < 1/4$, and $D$ is sufficiently large (i.e., in the lower bounds on $D$ in Theorem 3.1, if we replace $\delta$ by $3\delta$), we have from Theorem 3.1 that $\epsilon_{\text{unif-alg}}(m, 3\delta) > 1 - \epsilon$. Plugging these in Theorem J.1, we get the result in the above corollary. $\square$

### J.2 Deterministic PAC-Bayesian Bounds of Type B

In this section, we consider another standard approach to making PAC-Bayesian bounds deterministic [31, 22]. Here, the idea is to pick for each $h$ a distribution $Q_h$ such that for all $\mathbf{x}$:

$$\mathcal{L}^{(0)}(h(\mathbf{x}), y) \leq \mathbb{E}_{\tilde{h} \sim Q_h}[\mathcal{L}'^{(\gamma/2)}(\tilde{h}(\mathbf{x}), y)] \leq \mathcal{L}'^{(\gamma)}(h(\mathbf{x}), y),$$

where

$$\mathcal{L}'^{(\gamma)}(y, y') = \begin{cases} 0 & y \cdot y' \geq \gamma \\ 1 & \text{else.} \end{cases}$$

Then, by applying the PAC-Bayesian bound of Equation 28 for the loss $\mathcal{L}'_{\gamma/2}$, one can get a deterministic upper bound as follows, without having to introduce the extra $\Delta$ terms,

$$\mathcal{L}_{\mathcal{D}}^{(0)}(h) - \hat{\mathcal{L}}_S^{(\gamma)}(h) \leq \mathbb{E}_{\tilde{h} \sim Q_h}[\mathcal{L}'^{(\gamma/2)}(\tilde{h})] - \mathbb{E}_{\tilde{h} \sim Q_h}[\hat{\mathcal{L}}_S'^{(\gamma/2)}(\tilde{h})]$$
$$\leq \sqrt{2\epsilon_{\mathrm{pb}}(P, Q_h, m, \delta)} + 2\epsilon_{\mathrm{pb}}(P, Q_h, m, \delta).$$

We first define this technique formally:

**Definition J.2.** The distribution-dependent, algorithm-dependent, deterministic PAC-Bayesian bound of (the hypothesis class $\mathcal{H}$, algorithm $\mathcal{A}$)-pair is defined to be the smallest value $\epsilon_{\mathrm{pb\text{-}det\text{-}B}}(m, \delta)$ such that the following holds:

1. there exists a set of $m$-sized samples $\mathcal{S}_\delta \subseteq (\mathcal{X} \times \{-1, +1\})^m$ for which:
$$Pr_{S \sim \mathcal{D}^m}[S \notin \mathcal{S}_\delta] \leq \delta.$$

2. and if we define $\mathcal{H}_\delta = \bigcup_{S \in \mathcal{S}_\delta}\{h_S\}$ to be the space of hypotheses explored only on these samples, then there must exist a prior $P$ and for each $h$ a distribution $Q_h$, such that uniform convergence must hold as follows: for all $S \in \mathcal{S}_\delta$ and for all $h \in \mathcal{H}_\delta$,

$$\sqrt{2\epsilon_{\mathrm{pb}}(P, Q_h, m, \delta)} + 2\epsilon_{\mathrm{pb}}(P, Q_h, m, \delta) < \epsilon_{\mathrm{pb\text{-}det\text{-}B}}(m, \delta). \tag{35}$$

and for all $\mathbf{x}$:

$$\mathcal{L}^{(0)}(h(\mathbf{x}), y) \leq \mathbb{E}_{\tilde{h} \sim Q_h}[\mathcal{L}'^{(\gamma/2)}(\tilde{h}(\mathbf{x}), y)] \leq \mathcal{L}'^{(\gamma)}(h(\mathbf{x}), y) \tag{36}$$

as a result of which the following one-sided uniform convergence also holds:

$$\sup_{S \in \mathcal{S}_\delta} \sup_{h \in \mathcal{H}_\delta} \mathcal{L}_{\mathcal{D}}^{(0)}(h) - \hat{\mathcal{L}}_S'^{(\gamma)}(h) < \epsilon_{\mathrm{pb\text{-}det\text{-}B}}(m, \delta).$$

We can similarly show that $\epsilon_{\mathrm{pb\text{-}det\text{-}B}}(m, \delta)$ is lower-bounded by the uniform convergence bound of $\epsilon_{\mathrm{unif\text{-}alg}}$ too.

**Theorem J.2.** *Let $\mathcal{A}$ be an algorithm such that on at least $1 - \delta$ draws of the training dataset $S$, the algorithm outputs a hypothesis $h_S$ such that the margin-based training loss can be bounded as:*

$$\hat{\mathcal{L}}_S'^{(\gamma)}(h_S) \leq \hat{\epsilon}(m, \delta)$$

*and with high probability $1 - \delta$ over the draws of $S$, the generalization error can be bounded as:*

$$\mathcal{L}_{\mathcal{D}}'^{(\gamma)}(h_S) - \mathcal{L}_S'^{(\gamma)}(h_S) \leq \epsilon_{gen}(m, \delta)$$

*Then there exists a set of samples $\mathcal{S}_{3\delta}$ of mass at least $1 - 3\delta$, and a corresponding set of hypothesis $\mathcal{H}_{3\delta}$ learned on these sample sets such that:*

$$\left( \sup_{h \in \mathcal{H}_{3\delta}} \sup_{S \in \mathcal{S}_{3\delta}} \mathcal{L}_S^{(0)}(h) - \mathcal{L}_{\mathcal{D}}'^{(\gamma)}(h) \right) - (e^{3/2} - 1)(\hat{\epsilon}(m, \delta) + \epsilon_{gen}(m, \delta)) \leq \epsilon_{pb\text{-}det\text{-}B}(m, \delta).$$

Note that the above statement is slightly different from how Theorem J.1 is stated as it is not expressed in terms of $\epsilon_{\mathrm{unif\text{-}alg}}$. In the corollary that follows the proof of this statement, we will see how it can be reduced in terms of $\epsilon_{\mathrm{unif\text{-}alg}}$.

*Proof.* Most of the proof is similar to the proof of Theorem J.1. Like in the proof of Theorem J.1, we can argue that there exists $\mathcal{S}_{3\delta}$ and $\mathcal{H}_{3\delta}$ for which the test error can be bounded as,

$$\mathbb{E}_{\tilde{h} \sim Q_h}[\mathcal{L}_{\mathcal{D}}'^{(\gamma/2)}(\tilde{h})] \leq \mathcal{L}_{\mathcal{D}}'^{(\gamma)}(h) \leq \hat{\epsilon}(m, \delta) + \epsilon_{\mathrm{gen}}(m, \delta),$$

where we have used $\epsilon_{\mathrm{gen}}(m, \delta)$ to denote the generalization error of $\mathcal{L}'^{(\gamma)}$ and not the 0-1 error (we note that this is ambiguous notation, but we keep it this way for simplicity).

For convenience, let us denote by $a := \mathbb{E}_{\tilde{h} \sim Q_h}[\hat{\mathcal{L}}_S'^{(\gamma/2)}(\tilde{h})]$ and $b := \mathbb{E}_{\tilde{h} \sim Q_h}[\mathcal{L}_\mathcal{D}'^{(\gamma/2)}(\tilde{h})]$. Again, let us consider, for some $h \in \mathcal{H}_{3\delta}$ and $S \in \mathcal{S}_{3\delta}$, the case that $e^{3/2}b \geq a$. Then, we have, using the above equation,

$$
\begin{aligned}
\hat{\mathcal{L}}_S^{(0)}(h) - \hat{\mathcal{L}}_\mathcal{D}^{(\gamma)}(h) &\leq a - b \\
&\leq (e^{3/2} - 1)b \\
&\leq (e^{3/2} - 1)(\hat{\epsilon}(m, \delta) + \epsilon_{\text{gen}}(m, \delta)) \\
&\leq (e^{3/2} - 1)(\hat{\epsilon}(m, \delta) + \epsilon_{\text{gen}}(m, \delta) + \epsilon_{\text{pb-det-B}}(m, \delta)).
\end{aligned}
\tag{37}
$$

Now consider the case where $a > e^{3/2}b$. Again, by similar arithmetic manipulation in the PAC-Bayesian bound of Equation 28 applied on $\mathcal{L}'^{(\gamma/2)}$, we get,

$$
\begin{aligned}
\epsilon_{\text{pb}}(P, Q_h, m, \delta) &\geq a \underbrace{\ln \frac{a}{b}}_{\geq 3/2} - \frac{(a - b)}{2}\left(\frac{1}{(1-b)} + 1\right) \\
&\geq \frac{a - b}{2} \\
&\geq \frac{1}{2}\left(\mathcal{L}_S^{(0)}(h) - \mathcal{L}_\mathcal{D}'^{(\gamma)}(h)\right).
\end{aligned}
$$

Rearranging, we get:

$$
\begin{aligned}
\mathcal{L}_S^{(0)}(h) - \mathcal{L}_\mathcal{D}'^{(\gamma)}(h) &\leq \underbrace{2\epsilon_{\text{pb}}(P, Q_h, m, \delta)}_{\text{Applying Equation 35}} \\
&\leq \epsilon_{\text{pb-det-B}}(m, \delta).
\end{aligned}
\tag{38}
$$

Since, for all $h \in \mathcal{H}_{3\delta}$ and $S \in \mathcal{S}_{3\delta}$, one of Equations 37 and 38 hold, we have the claimed result.

$\square$

Similarly, as a result of the above theorem, we can show that $\epsilon_{\text{pb-det-B}}(m, \delta) = \Omega(1) - \mathcal{O}(\epsilon)$, thus establishing that, for sufficiently large $D$, even though the generalization error would be negligibly small, the PAC-Bayes based bound would be as large as a constant and hence cannot explain generalization.

**Corollary J.2.1.** *In the setup of Section 3, for any $\epsilon, \delta > 0, \delta < 1/12$, when $D = \Omega\left(\max\left(m \ln \frac{3}{\delta}, m \ln \frac{1}{\epsilon}\right)\right)$, we have,*

$$
1 - (e^{3/2} - 1)\epsilon \leq \epsilon_{\text{pb-det-B}}(m, \delta).
$$

*Proof.* It follows from the proof of Theorem 3.1 that $\hat{\epsilon}(m, \delta) = 0$, since all training points are classified by a margin of $\gamma$ (see Equation 9). Similarly, from Equation 12 in that proof, since most test points are classified by a margin of $\gamma$, $\epsilon_{\text{gen}}(m, \delta) \leq \epsilon$. Now, as long as $3\delta < 1/4$, and $D$ is sufficiently large (i.e., in the lower bounds on $D$ in Theorem 3.1, if we replace $\delta$ by $3\delta$), we will get that there exists $S \in \mathcal{S}_{3\delta}$ and $h \in \mathcal{H}_{3\delta}$ for which the empirical loss $\mathcal{L}^{(0)}$ loss is 1. Then, by Theorem J.2, we get the result in the above corollary. $\square$