[Reviews · NeurIPS 2019]

Reviewer 1



This paper emphasizes that uniform convergence is inherently problematic for explaining generalizaiton of deep learning, as even the algorithm-dependent application would fail – casting doubt on post-Zhang et al. [31] algorithm-dependent approaches. The whole paper is well written and the related results are important.

Reviewer 2



The paper is clearly written, and the claim of the title is original. However I don't think that the claim is correct.

Reviewer 3



== SUMMARY == There has been much recent work that tries to explain why over-parameterized neural networks can generalize to unseen data, when classical learning theory says that they should not. This paper argues that the problem is uniform convergence, that there are certain scenarios in which it will fail to yield meaningful bounds, even when the learned model does generalize. The key observation is that the norm of the learned weights can grow with the number of examples. The paper demonstrates this empirically, then provides a theoretical construction that proves it for high-dimensional linear models. It also conducts an experiment to support the claim for 1-layer ReLU networks. The idea that uniform convergence bounds are inadequate for explaining generalization in some models/algorithms is not new. For example, it's well known that the 1-NN classifier has infinite VC dimension. But, it is surprising that the weight norm can grow with the number of examples. Then again, the construction that's used to show this is a bit of a "straw man." It doesn't use any regularization, the whole point of which is to constrain the norm of the weights. So, the paper has shown that uniform convergence fails to explain generalization in pure ERM, but it says nothing about SRM (structural risk minimization). I guess what I'm trying to say is that, while there _exist_ algorithms that can generalize for which uniform convergence fails, there are other algorithms that may generalize just as well for which uniform convergence _does_ hold. I appreciate the conclusion that there should be more research into algorithmic stability, but I don't think it's fair to say that it has "not been as extensively studied for deep networks." Hardt et al.'s paper (2016) sparked a lot of follow-up work. I think the big question in stability analysis is whether the common assumptions and constants (e.g., the Lipschitz constant of the network) are reasonable in deep learning. I enjoyed reading this paper. I like that it has a bold message that calls into question a lot of recent work. It's very well written, clear and easy to follow. The theoretical results are stated cleanly, with minimal notation. The experimental methodology is explained fairly well. I appreciate that the technical proofs were deferred, but that a proof sketch was provided. == DETAILED COMMENTS == For what it's worth, the construction used to prove Theorem 3.1 is basically Perceptron, only with updates on correct predictions too. Based on the way equations are typeset (especially in the appendix), it looks like the paper was originally written for a 2-column format. I suggest getting rid of some of the line breaks to take advantage of the single column format. The spade notation used in line 150 is fun, but seems a bit out of left field. Line 169: needs punctuation between "bounds namely". Line 251: "initialized to origin" --> "initialized to the origin". == POST AUTHOR RESPONSE == My main concern was that analyzing pure ERM (i.e., with no regularization) is a bit of a straw man. The author response reminded me that many prior works have studied this setting, and that it is common to train DNNs that generalize with no regularization. I am therefore raising my score to 8. That said, I would still like to see more discussion of regularization in the paper -- at the very least, an acknowledgement that capacity control would enable generalization with uniform convergence guarantees. It would be interesting to analyze learning with very small amounts of regularization (e.g., lambda = 10^{-5}) -- which is also a very common scenario -- and see whether uniform convergence with meaningful rates is possible there. Lastly, I do hope that the authors amend the paragraph about stability research in the conclusion. As written, it minimizes an extensive body of work.

Reviewer 4



Originality: I have seen a presentation/workshop paper on the same topic (I believe by the same authors) in the ICML2019 workshop on generalization in deep learning. Other than that, I am not familiar with earlier work presenting similar counterexamples showing UC cannot explain generalization. Clarity: The paper is mostly clear and easy to read. Quality: The proofs of the main theoretical results are correct. The results seem to be quite useful and informative. I believe that the paper is of relatively high quality. There was one point of confusion for me regarding Pseudo-Overfitting. Clarifying this concept and the framing of counterexamples presented with respect to this concept would be required (but possibly not sufficient) for me to increase my ranking to "Top 15%". From my understanding the counterexamples provided *do* pseudo-overfit, while the paper mentions (and purports to demonstrate in appendix B) that deep networks do not pseudo-overfit. This confusion casts doubt on the relevance of the counterexamples to actual models used in practice. I think that this could be clarified further by the authors. Significance: The work is significant as it informs theorists that certain tools may not be useful in their research. This is significant, IMO, to warrant acceptance at NeurIPS. It does not, however, provide an alternative set of tools for theorists to use when UC is not necessary. This would be required for me to consider this a "Top 5% paper". I imagine that the counterexamples presented in this work could become "standard text book examples" in the future since they are clear and easy to follow, and also readily demonstrate the phenomenon of interest.

[Author Response · NeurIPS 2019]

We thank the reviewers for their feedback, especially R1, R4 and R5 for appreciating the significance of our work at
this juncture in deep learning theory. Below, we respond to the reviewers in order.

**R1**: Again, we thank the reviewer for their encouraging comments. As indicated in their suggested improvements, we
will certainly add a table that lists all existing uniform convergence bounds, in future versions of the paper.

**R3**: Based on our understanding, R3 believes that our negative result about the overparameterized linear model in Thm
3.1 – although correct – is trivial because it is already known that "uniform convergence (u.c.) holds only when dataset
size is proportionally larger than dimension". This is a strong claim which implies that u.c. is known to not hold in *any*
overparameterized linear setting (hence implying Thm 3.1). **First**, there is no such strong statement in learning theory.
(Nor is there a specific result like ours which shows that even the *tightest* u.c. bound, namely $\epsilon_{\text{unif-alg}}$, can fail for *some*
overparameterized linear models). **Second**, **this strong claim is incorrect as it contradicts fundamental results** like
margin-based u.c. bounds for SVMs (Theorem 4.4. in [18]), which are known to be meaningful even in infinite
dimensions. **Third**, we must emphasize our ReLU example in Sec 3.2, which is not mentioned in the review. This
example is nearly identical in terms of its "dimensionality" to common settings like MNIST (parameter count $\gg$ dataset
size, input dimensionality is $\approx 10^3$, the dataset size is $\approx 50k$). If it is indeed trivial that for these dimensions & dataset
size, u.c. would fail, then it follows immediately that the u.c. bounds proposed for these settings by the dozens of
post-Zhang-et-al. papers, are all *obviously* pointless – which is clearly not the case. **Finally**, we'd also like to draw
R3's attention to the empirical contributions in Sec. 2 that have not been mentioned in the review. These are new and
constitute half the paper. Keeping these facts in mind, we politely request a fair and complete re-evaluation of the paper.

**R4**: We thank the reviewer for their suggestions, and also for stating their main concern precisely. R4 believes our
examples are somewhat a strawman because they show failure of uniform convergence (u.c.) only in cases without
explicit regularization. We must strongly emphasize that **this is not a strawman: our unregularized setting is in**
**fact, the precise setting that has been the focus of Zhang et al., '17 and all the other dozens of follow-up work.**
The key surprising phenomenon in deep learning that has gained significant theoretical interest is the fact that deep
networks generalize *even when there's no explicit control*, either on the parameter count or on the norms – the lack of
regularization is pivotal to the "surprise" here. To this end, post-Zhang et al., works developed u.c. bounds with the goal
of explaining generalization in this *unregularized* setting – and this goal has been elusive. Our work is a warning that
this particular active, ongoing pursuit may after all be a futile exercise unless we go beyond u.c. Finally, indeed, the
reviewer is right in noting that u.c. may still hold in other settings (with regularization, compression, SRM etc.,). These
settings, however, are somewhat orthogonal to the main generalization puzzle (and we make no claims about these
settings). To conclude, we hope our response explains why our examples are certainly not a strawman in the context of
current deep learning theory research, and thus, we hope this helps in re-evaluating the paper.

**R5**: We thank R5 for their careful reading & thorough summary. The main concern of R5 is that, while we claim
deep networks "do not suffer from pseudo-overfitting" (by which we mean, the gap between the mean test and training
margins of deep networks does decrease with training data size), it seems that our examples do suffer from pseudo-
overfitting. Hence, R5 is wary of the relevance of our examples to deep learning. This is an interesting point, and we
argue why this is actually not of concern. And we will certainly add the following discussion to the paper.

**First**, in our hypersphere example, as shown in the accompanied figure, the mean
margins on the test data (orange line) and on training data (blue line) do converge
to each other with more training data size $m$ i.e., **the gap in the mean test and**
**training margins (green line) does decrease with $m$**. Thus our setup exhibits a
behavior similar to deep networks on MNIST in Figure 11 in our paper. As noted in
lines 565-570, since the rate of decrease of the mean margin gap in MNIST is not as
large as the decrease in test error itself, there should be "a small amount" of psuedo-
overfitting in MNIST. The same holds in this setting, although, here we observe an
even milder decrease, implying a larger amount of pseudo-overfitting. (Nevertheless,
uniform convergence cannot capture even this decrease with $m$.) **Secondly**, we
must note that here we train an actual ReLU-based network using vanilla SGD just
like in the MNIST example. Hence, if any bound claims to "explain generalization

in deep learning", it *should* explain generalization in our example – and we establish that uniform convergence bounds
cannot do the trick. This makes our example relevant to deep learning regardless of pseudo-overfitting. **Third**, the
reviewer is certainly right in that our linear example does suffer from pseudo-overfitting. However, we must emphasize
that this psuedo-overfitting in itself does not imply Theorem 3.1's lower bound on $\epsilon_{\text{unif-alg}}$ as noted in lines 565-570
(as pseudo-overfitting implies a lower bound only on a specific class of uniform convergence bounds). We do think
that pseudo-overfitting is a phenomenon worth exploring better; however, we also believe that there is a phenomenon
beyond pseudo-overfitting that is at play in deep learning, which our examples elucidate. We hope these three points
inform the reviewer as to why pseudo-overfitting is not of concern while drawing implications from our examples.

[Meta-Review · NeurIPS 2019]

The paper initially received two strong reviews and one very negative review. I solicited the feedback from an additional reviewer who read the paper thoroughly and agreed with the first two reviewers. I believe this paper makes a valuable contribution to the study of generalization in modern machine learning settings. In particular, they appear to have isolated ways in which the learned classifier inherits microscopic structure from the data that causes it to misclassify a ghost data set distributed identically to the training data. The result is catastrophic for two-sided generalization error. I suspect this paper will generate much downstream insight.